# The ISG Atlas: a loss-of-function analysis characterizes antiviral properties of interferon stimulated genes

Karsten Krey [1,7], Jennifer Risso-Ballester [1,7], Sabri Hamad [1], Susanne Maidl[1], Sara Bilekova [2,3,4], Quirin Emslander[1], Melissa Verin[1,5], Sarah Mundigl[1], Alexandrina Cernat[1], Antonio Piras[1], Valter Bergant [1], Vincent Grass [1] & Andreas Pichlmair [1,5,6] ✉

The innate immune system requires the activity of interferon-stimulated genes (ISGs) to mount its protective response against viruses. However, the activity of ISGs against viruses varies widely and is orchestrated by the interplay of hundreds of ISGs. Utilizing a time-resolved, arrayed loss-of-function screen, we systematically investigate 285 ISGs for their virus-modulating activity against eight viruses. The quantitated data from the screen results do not necessarily result in similar quantitative biological effects of gene function but indicates virus specificity of many ISGs and pan-proviral activity of some ISGs, such as RNA 2′,3′-cyclic phosphate and 5′-OH ligase (RTCB). Co-depletions of selected candidates identify ISGs with synergistic functions, highlighting particularly strong synergies between ISGs inhibiting entry pathways and ISGs involved in IFN signaling. Among unexplored ISGs, we identify BORCS8, which has a particularly prominent role in modulating SARS-CoV-2 infection. Mechanistically, BORCS8 mediates the acidification of early endosomes during viral entry, a process known to facilitate the degradation of virus particles. Collectively, this extensive resource reveals specificities of ISGs identified in this screening system and suggests potential strategies for antiviral treatment options.

The antiviral innate immune response, orchestrated by the interferon (IFN) system, serves as a primary defense against viral infections. Type-I IFNs play a pivotal role in this system by modulating the expression of IFN-stimulated genes (ISGs), which serve as the downstream effectors bearing diverse antiviral properties[1–3]. Up to 10% of all human genes, covering a wide diversity of processes, are regulated by IFNs[3]. Functionally, some ISGs directly intervene with specific stages of the viral life cycle or modulate general cellular processes, such as protein stability, translation, and cell death, thus mediating indirect inhibition of virus replication. Viruses have evolved highly diverse strategies to counteract host defenses, including the targeting of IFN signaling and direct targeting of ISGs[4]. To maximize the efficacy of the cellular defense system, the host has evolved a large repertoire of ISGs with diverse functions, resulting in a broad array of tools to effectively control a wide diversity of viruses. The constant tug-of-war between virus and host has resulted in staggering complexity and robustness of the IFN system and its ISGs[5].

[1]Institute of Virology, School of Medicine and Health, Technical University of Munich, Munich, Germany. [2]Institute of Diabetes and Regeneration Research (IDR), Helmholtz Diabetes Center, Munich, Germany. [3]German Center for Diabetes Research (DZD), Neuherberg, Germany. [4]Technical University of Munich, School of Medicine, Munich, Germany. [5]Systems Virology, Institute of Virology, Helmholtz Center Munich, Munich, Germany. [6]German Center for Infection Research (DZIF), Munich Partner Site, Munich, Germany. [7]These authors contributed equally: Karsten Krey, Jennifer Risso-Ballester. ✉e-mail: andreas.pichlmair@tum.de

The IFN system is under control of several pattern recognition receptors (PRRs), including DDX58 (RIG-I), MDA5, and cGAS[6]. It requires signaling cascades involving interferon regulatory factors (IRF3, IRF7, and IRF9) and signal transducers and activators of transcription (STAT1 and STAT2), which are collectively essential to regulate expression of ISGs to establish an effective antiviral state[7,8]. While some ISGs counteract processes commonly used by viruses and thereby target many different pathogens, other ISG functions are more specific and only effective against certain viruses or virus families. Among the best-characterized direct-acting ISGs are MX proteins, which bind viral nucleoproteins or capsids to exert antiviral activity. Similarly, IFIT proteins bind to specific viral RNA features to perturb their functionality[9]. While MX and IFIT proteins exhibit significant antiviral activities against specific pathogens, they are ineffective against others or even proviral[10–12]. Similar virus-specificities have also been reported for other ISGs, such as IFITMs, TRIM5α, and CH25H.

To characterize the antiviral properties of ISGs, several over-expression approaches have been employed. These seminal reports were instrumental in characterizing the activity of key ISGs, such as IRF1, C6orf150, DDX58, MDA5, and cGAS, as pan-antiviral factors[13,14]. Moreover, virus-specific effects were also characterized by this approach, e.g., the antiviral activity of ISG20 against Bunyaviruses[15], CCDC92 against Ebola virus[16], and LY6E against SARS-CoV-2[17]. However, the exogenous expression of some ISGs is not tolerated by the cell, and the expression of some individual ISGs is not sufficient to elicit antiviral functions, which complicates their study by gain-of-function screens. Recently, the use of loss-of-function studies has complemented the gain-of-function approach. Gene depletion screening techniques were invaluable for uncovering genes necessary to viral replication or characterizing novel viruses like SARS-CoV-2[18–20]. While pooled genome-wide screens are frequently employed due to their high throughput and straightforward experimental design[21], they come with limitations, such as susceptibility to paracrine influences within the cell population. For instance, pattern recognition receptors (PRRs) and associated signaling molecules play a crucial role in inducing the production of antiviral cytokines. However, they are not commonly identified in pooled loss-of-function screens since cytokines generated by other cells within the population compensate for the loss of PRRs and PRR signaling molecules. In contrast, the arrayed depletion of individual genes offers enhanced sensitivity due to a clear genotype-to-phenotype relationship and allows for a more nuanced quantification of effects[22].

To gain insight into the virus-specific activities of ISGs we here employ an arrayed loss-of-function screen to elucidate the antiviral activities of 285 ISGs against a panel of eight viruses. We utilize a live-cell imaging platform to track the infection of fluorescent reporter viruses, providing a time-resolved quantification of virus infection dynamics for each ISG depletion. The quantitative data obtained through the screening results may not follow similar quantitative biological effect sizes. However, this dataset reveals novel antiviral specificities and pan-antiviral functions of ISGs, gives insights into the overall organization of the innate immune response and suggests options that may be explored for antiviral therapies in the future.

## Results

### Functional assessment of ISG–virus interactions

We established a live-cell imaging screening system to study the virus-modulating activities of ISGs in A549 cells, which are widely used in virus research and are well-studied for their responsiveness to type-I IFNs[23]. The cells were engineered to stably express eGFP or mRFP-tagged histone H2B to enable a fluorescence-based read-out as a proxy for the cell number (Fig. 1a). We selected eight different fluorophore-expressing viruses belonging to different virus classes: Herpes simplex virus 1 (HSV-1) strain 17 + , Measles virus (MeV) strain EdTag, Rift Valley Fever Virus lacking Non-structural protein S (RVFVΔNSs) ZH548 strain,

Severe acute respiratory syndrome coronavirus 2 (SARS-CoV-2) MUC-IMB-1 strain, Semliki Forest virus (SFV) SFV6 strain, Vaccinia virus (VACV) WR strain (V300), Vesicular stomatitis virus (VSV) Indiana strain, and Yellow Fever virus (YFV) 17D strain and determined the half-maximal effective concentration (ED50) of IFNα for each virus (Supplementary Fig. 1a).

Based on data from the Interferome database[24], we selected 300 genes, of which 232 were shown to be up- (ISG) and 68 down-regulated (IFN-repressed gene, IRG) in response to type I IFNs (Supplementary Data 1). Henceforth, "ISG" will denote both ISGs and IRGs. We then stably depleted these ISGs in A549 cells using CRISPR/Cas9-based lentiviral vector pools (Fig. 1a, Supplementary Fig. 1b, c), where each gene was targeted by three sgRNA sequences selected for high on-target and low off-target scores, as well as low self-complementarity. Knockout (KO) efficiency was tested for a subset of ISGs using RT-qPCR. All tested KO cells showed significant transcript reductions for the targeted ISGs (Supplementary Fig. 1d). To avoid experimental bias, we excluded 15 ISG KO cells, which showed reduced cell growth by more than 25% in an observation period of 72 h (Supplementary Fig. 1e, Supplementary Data 2). To study the specific effect of the remaining 285 ISGs on virus infection dynamics, we induced an ISG response by pre-stimulating the KO and non-targeting control (NTC) cell lines with IFNα for 6 h prior to infection. All cell lines were then infected with the selected reporter viruses, and fluorescence images were acquired in 2–3 h intervals for 48–72 h (Fig. 1a). In total, we generated 730,802 epifluorescence images capturing the virus infection kinetics in relation to the depletion of individual ISGs. We calculated the ratio of the virus-expressed fluorophore signal and the H2B-fluorophore signal, providing a normalized measure of virus-associated reporter signal relative to the cell number (Supplementary Fig. 2a, b). We fitted this ratio to a logistic growth curve and normalized all curve parameters to the corresponding parameters of the NTC samples. The parameters describing a sigmoidal curve are the time-independent upper asymptote ($K$-value; representing the maximum signal reached during the observation period), the time until K/2 was reached ($\tau$-value; representing time post infection until the half maximal signal is reached), and the slope at t = $\tau$ ($\beta$-value; representing the increase of virus growth at one timepoint) (Fig. 1b, Supplementary Data 2). Expression of H2B-fluorophore data indicated relative homogeneous expression, ensuring that the normalization of the virus-associated reporter signal to the cell number does not bias this analysis (Supplementary Fig. 2c). To accommodate the varying dynamic ranges of the reporter viruses, we employed z-scoring to normalize the parameters within each virus dataset. The parameters of viral replication curves for the NTCs closely aligned with the mean of all tested ISGs for most viruses, except for HSV-1, for which the NTC curves consistently showed reduced values for an unknown reason (Supplementary Fig. 2d, e).

To comprehensively illustrate the resulting data, we considered $K$- and $\tau$-values, as well as the cell growth rate, for all targeted ISGs and viruses. We plotted these values on a heatmap using hierarchical clustering of $K$-values to illustrate the complex and nuanced activity of ISGs and to reveal pan-proviral and pan-antiviral activities, as well as virus-specific activities of individual ISGs (Fig. 1c, Supplementary Data 2). The IFNα signaling components STAT1, IRF9, and the translation regulator EIF2AK2 (PKR) are well-studied for their antiviral function against diverse viruses as well as their ability to regulate innate immune signaling[1,7,8] and were among the ISGs with broad antiviral activities. Moreover, our data confirmed the broad antiviral activity of OAS3, a main activator of the OAS/RNase L signaling pathway[25]. Importantly, our analysis also correctly identified ISGs with broad antiviral functionality, such as DDX58, which predominantly exhibited antiviral activities against RIG-I-dependent RNA viruses[26], while it did not show an effect for HSV-1 or SARS-CoV-2. Our data also confirmed the antiviral activity of IFI16 against HSV-1 and VACV but, in addition, highlighted its role in SARS-CoV-2 replication, which is in line

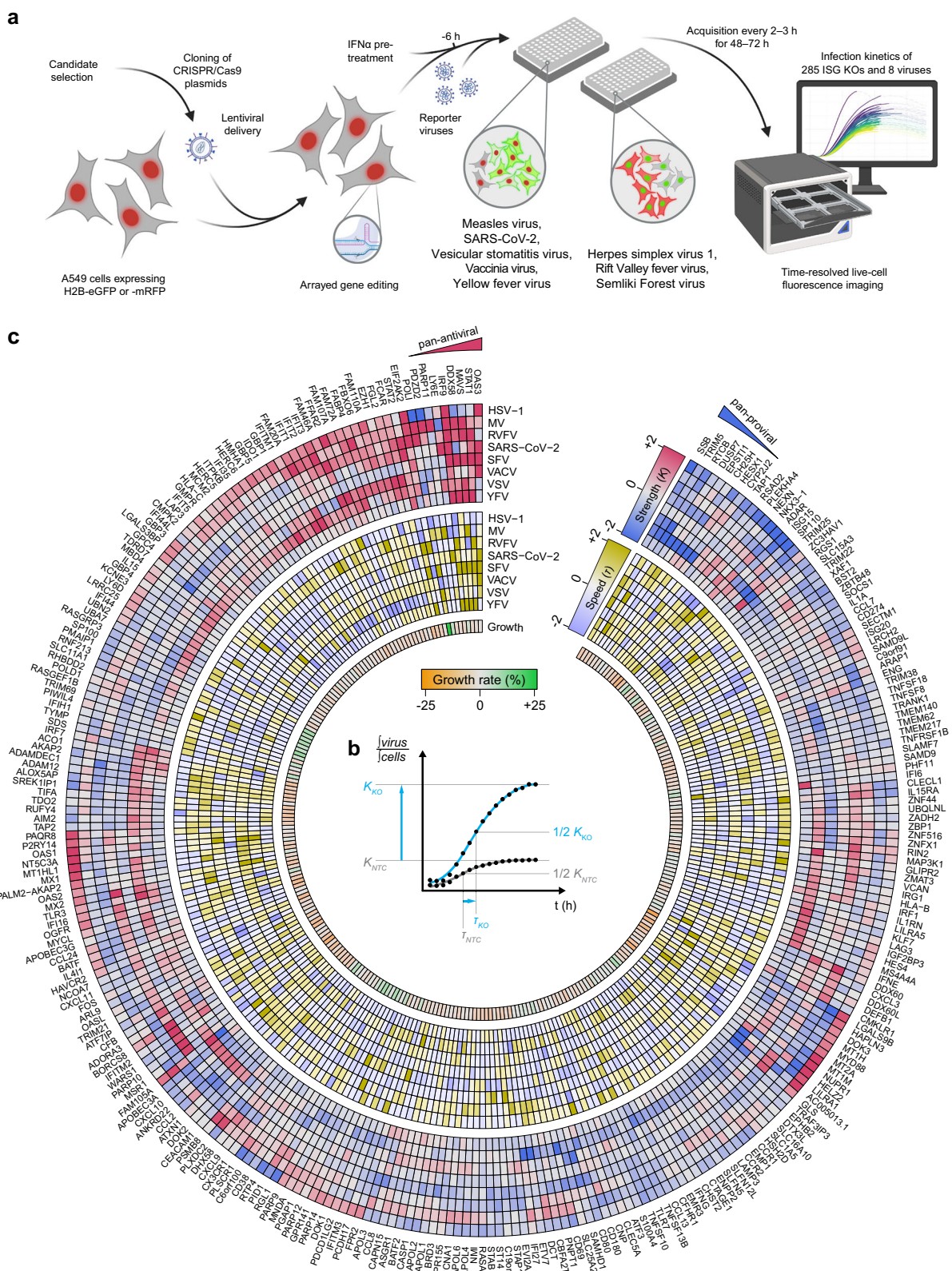

with recent reports indicating that IFI16 plays a broad role in antiviral immunity for some viruses[27–29]. Notably, ISGs that are known to be functionally connected, i.e., by operating in similar pathways (e.g., STAT1 and IRF9[7,8]) or co-dependencies (e.g., IFIT1, −2, −3−which form hetero complexes to exhibit their full activity[30]) clustered together in this analysis, further indicating the validity of this approach and the value of this dataset (Fig. 1c). The identification of known activities of

well-characterized ISGs supports the correct and specific CRISPR/Cas9-mediated targeting and the validity to assess virus growth used in this study.

Some ISGs displayed surprisingly broad antiviral activities. These included FBXO6, an E3 ligase involved in ER-associated protein degradation (ERAD)[31]. FBXO6 has previously been shown to regulate immune responses and cell death induction by influenza A virus

**Fig. 1 | Functional assessment of ISG–virus interactions. a** Schematic representation of the experimental setup. A549 cells stably expressing either H2B-mRFP or H2B-eGFP were targeted for the depletion of 300 ISGs using three CRISPR/Cas9 gRNA sequence templates by lentiviral delivery. Knockout (KO) cells showing a reduction in cell growth of more than 25% compared to non-targeting control (NTC) cells were excluded. The remaining 285 ISG KO cells were pre-treated with IFNα at a virus-specific dose for 6 h and infected with eight reporter viruses. Each virus expresses a fluorophore complementary to the color of the H2B fluorophore in the cells (green virus-expressed fluorophore on H2B-mRFP and vice versa). The progression of the viral infection was captured using live-cell fluorescence imaging. Created in BioRender. Pichlmair, A. (2026) https://BioRender.com/wp5zo4e. **b** The ratio of the integrated virus-expressed fluorophore signal to the integrated H2B-fluorophore signal provides a normalized measure of virus infection dynamics relative to the cell number. This ratio was fitted to a logistic growth curve, allowing the quantification of key infection dynamics parameters: infection strength (K-value, upper asymptote) and the time to reach K/2 (τ-value). **c** Hierarchical clustering based on K-values showing the modulation of virus infection for each ISG depletion. K-values (red-to-blue), τ-values (yellow-to-blue), and the growth rate of uninfected cells (green-to-orange) are shown. All values are normalized to their respective NTC values and z-scored per virus. Shown is the mean of $n = 3$ biological independent replicates. Values are capped from $|z| > 2$.

through degradation of NLRX1[32] and IFN type-I-mediated antiviral immunity induced by VSV, Sendai virus and synthetic ligands[33]. However, its broad antiviral activity has not yet been described. Furthermore, we identified an unexpected broad antiviral activity of FAM105A (OTULINL), a pseudo-deubiquitinase lacking deubiquitinase activity with a yet undescribed function[34]. Notably, while FAM105A has broad antiviral activity against HSV-1, MeV, RVFV, and VACV, it was proviral for VSV. Moreover, FAM20A, a poorly defined pseudokinase, exhibited similarly unexpected broad antiviral activity. Proteins linked to transcription, like the histone-lysine N-methyltransferase EZH1, as well as FAM46A (TENT5A)—a cytoplasmic non-canonical poly(A) RNA polymerase—also showed antiviral activity against multiple viruses. For FAM46A, we could also recapitulate previously reported inhibitory activity for MeV and VACV[14]. Similarly, FAM72A, a protein potentially involved in regulating cellular reactive oxygen species metabolism and cell growth[35], was active against 5 of the 8 tested viruses.

Besides antiviral ISGs, we could identify several ISGs with proviral activity across different viruses. These ISGs included SSB, TRIM5, and RTCB. TRIM5—an E3 ubiquitin ligase that blocks reverse transcription of human immunodeficiency virus-1 (HIV-1)[36,37]—exhibited particularly prominent proviral activity for VACV and VSV within this dataset. The RNA chaperone La/SSB (SSB) has previously been shown to bind an internal ribosome entry site (IRES) of the Hepatitis C virus (HCV), a key component of virus translation, thereby limiting viral replication[38,39]. Surprisingly, our data suggests that, besides inhibiting HCV, SSB also exhibits proviral activities for HSV-1, RVFV, VACV, and VSV, indicating additional roles of SSB beyond regulating IRES-dependent translation. A similar dichotomy in virus-specific effects was observed for several other ISGs, including POLI, FAM105A, MSR1, and ZC3HAV1. Polymerase iota (POLI), involved in DNA repair[40], restricted MeV and RVFV, but was required for the growth of HSV-1 and VSV. FAM105A showed antiviral effects against HSV-1, MeV, RVFV, and VACV, but was required for VSV. Similarly, the macrophage scavenger receptor 1 (MSR1) exhibited antiviral activity against SARS-CoV-2 and VACV but was found to be proviral for MeV. Another example is ZC3HAV1, which was proviral for VACV but restricted SFV and VSV replication. ZC3HAV1 has previously been shown to be a restriction factor for modified vaccinia virus Ankara (MVA)—an attenuated strain of VACV—but its depletion had no effect on VACV propagation[41]. Since our screen suggests that ZC3HAV1 serves as a prominent host factor for VACV propagation, an additional in-depth evaluation of ZC3HAV1's activity in the context of poxviruses may be necessary.

The antiviral activities of ISGs were systematically tested using gain-of-function approaches[13,14,16,42–44]. Our selection of ISGs contained 111 ISGs, which were also evaluated in previous overexpression screens[13,14,45]. We intersected our loss-of-function data with gain-of-function studies for MeV, VACV, and YFV (Supplementary Fig. 3a). Notably, we observed that the gain-of-function screens identified a different subset of ISGs with antiviral activity, as compared to ISGs identified by the loss-of-function approach used here. The depletion of OAS3, for instance, enhanced the propagation of VACV, its over-expression had a negligible effect. Similarly, the overexpression of IRF1 robustly inhibited the replication of YFV, while the KO only had a minor

effect. ISG overexpression and depletion thus serve as complementary methods for investigating cellular mechanisms, illustrating the added value of both strategies.

To evaluate these findings in different cell types, we depleted a subset of selected ISGs in SK-N-SH neuroblastoma cells and primary human foreskin fibroblasts (HFF) and challenged them with RVFV, HSV-1, and VACV (Supplementary Fig. 3b, c). These experiments showed that the majority of ISG functions were conserved between different cell types, but we could also identify virus- and cell-type-specific activities. For HSV-1, for instance, antiviral functions were similar in all tested cell lines, but the magnitude of the response was generally lower in SK-N-SH cells and HFFs. Some individual ISGs appear to differ in their antiviral properties among the tested cell types. The antiviral effect of NUPR1 against VACV, for instance, was particularly prominent in HFFs as compared to other tested cell types in this screen. Notably, some ISGs had cell-type-specific functions. An example was HELZ, which was proviral for VACV in A549 cells but antiviral in HFFs and SK-N-SH cells. Similarly, DDX60 and MX2 had opposing effects in RVFV-infected A549 and HFFs. This analysis shows that the majority of ISGs had similar activities in different cell types, but that a subset of ISGs have more pronounced or nuanced effects in specific cell types, which is in line with other reports on cell-type specificity of ISGs[1,46–49].

Collectively, our data revealed pro- and antiviral properties of a large number of ISGs with yet unreported functions. While a relatively small subset of ISGs was broadly active against many viruses, we identified numerous ISGs that showed virus-specific effects.

## Delineating ISGs' specificity to modulate virus growth

The depletion of most ISGs had minimal or no impact on most tested viruses (Fig. 2a). A few ISGs per virus displayed major effects ($|z| \geq 2$ standard deviations (SDs)), while many ISGs appeared to only subtly regulate virus infection ($0.5 \leq |z| < 2$ SDs). Approximately one quarter of the tested ISGs displayed pro- and antiviral activity against individual viruses. This suggests that the combined activities of multiple ISGs contribute to the total antiviral activity against individual viruses. The most robust antiviral activities per virus were observed for EIF2AK2 (HSV-1), PARP11 (MeV), IRF9 (RVFV and YFV), LY6E (SARS-CoV-2), OAS3 (VACV), and IFITM1 (VSV) (Fig. 2b). Lymphocyte antigen 6E (LY6E) was identified to be active against SARS-CoV-2, which aligns with its known role in restricting SARS-CoV-2 entry[17,50]. Moreover, correctly identifying well-studied ISGs such as IRF9, EIF2AK2, OAS3, and IFITM1 among the most prominent hits[1] validated our screening approach. This analysis also identified several ISGs with yet unreported antiviral activity, like POLI and PDZD2—a protein similar to pro-interleukin-16[51]—which showed antiviral activity against MeV. The same analysis also allowed us to identify ISGs with specific virus-promoting functions. These included LGALS9B for MeV and RVFV, ATXN1 for SARS-CoV-2, ADAMDEC1 for SFV, RTCB for VSV, GPR141 for YFV, POLI for HSV-1 and ZC3HAV1 for VACV.

For τ-values (time to reach K/2), we observed a similarly large subset of ISGs altering the infection progression (Fig. 1c, Supplementary Fig. 3d). Among these, the candidates causing the most

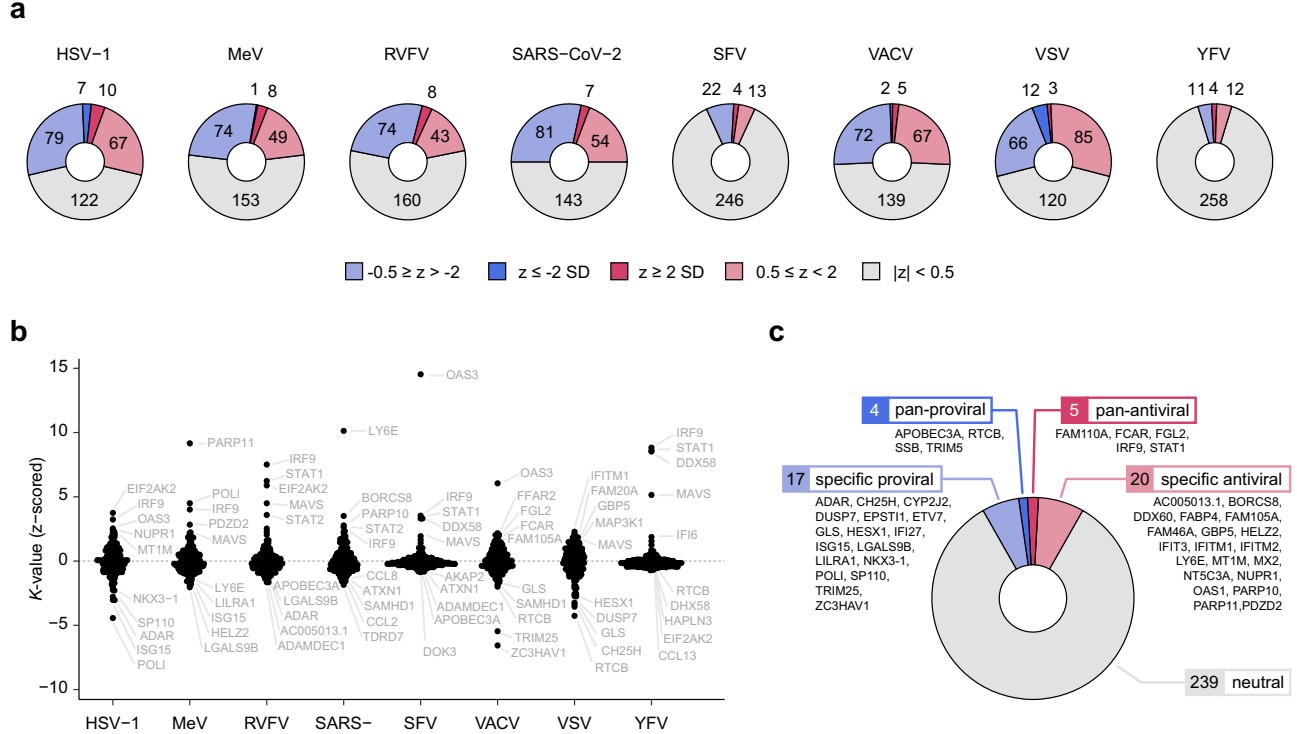

**Fig. 2 | Delineating ISGs specificity to modulate virus replication. a** Categorical effects of ISG knockouts (KOs) based on their $K$-values: neutral (gray), minor pro- or antiviral ($0.5 < |z| \leq 2$ SDs, light blue and light red, respectively), and major pro/antiviral ($|z| \geq 2$ SDs, blue and red, respectively). SD: standard deviation. **b** Beeswarm plots of $K$-values after z-scoring, showing all assessed KO effects. **c** Concatenated ISG effects with respect to $K$-values across all viruses, categorizing ISG KOs as virus-specific, pan-antiviral, pan-proviral, or neutral. ISGs showing major effects against a single virus but only minimal or no effects for the other viruses were considered virus-specific. ISGs exhibiting pro- or antiviral effects among at least six of the eight tested viruses were considered pan-proviral or pan-antiviral, respectively.

pronounced delay in half maximum infection signal were RIN2 (HSV-1), PARP11 (MeV), STAT2 (RVFV), LY6E (SARS-CoV-2), OAS3 (SFV and VACV), OAS1 (VSV), and IRF9 (YFV) (Fig. 1c, Supplementary Fig. 3e). On the other hand, candidates causative for an accelerated time to K/2 were POLI (HSV-1), HELZ2 (MeV), MAVS (RVFV), CXCL11 (SARS-CoV-2), EIF2AK2 (SFV), ZC3HAV1 (VACV), IFI16 (VSV), and IFI35 (YFV).

To concatenate the data more comprehensively, we combined $K$-values from all viruses to identify ISGs with consistent pro- or antiviral effects among at least six of the eight viruses tested (Fig. 2c). When considering both minor and major effects, five candidates—FAM110A, FCAR, FGL2, IRF9, and STAT1—were broadly antiviral. At the same time, APOBEC3A, RTCB, SSB, and TRIM5 were pan-proviral. This multi-virus analysis revealed key ISGs, such as LY6E and POLI, with solid effects on the replication of specific viruses while highlighting novel pan-anti and proviral ISGs.

## A machine-learning approach to identify functional relationships

To further investigate the complex relationship between ISGs and the tested viruses, we employed an unsupervised machine-learning approach. Self-organizing maps (SOM) iteratively learn the underlying data structure in a multi-dimensional space, represented by the z-scored input parameters—$K$-, $\tau$-, and $\beta$-values for all candidates and viruses—and assign a categorical neuron to each ISG (Fig. 3a, Supplementary Fig. 4a). Because the SOM is trained on z-scored parameters, it primarily captures relative patterns and correlations across viruses rather than absolute effect magnitudes. The distribution of ISGs across the SOM revealed intriguing patterns of virus-host interactions (Fig. 3b). For instance, some neurons contained many ISGs with only minor effects on viral infection (Fig. 3b; e.g., neurons 11, 16), while

other neurons contained few ISGs with dominant effects (Fig. 3b; neurons 2, 3, 5, 6, 12, 13, 17, 20). Notably, many ISGs with similar functions were grouped into the same neuron. This is exemplified by proteins involved in type-I IFN signaling, such as DDX58, IRF9, MAVS, STAT1, and STAT2, all assigned to neuron 5. This approach also revealed some unexpected similarities between ISGs. In neuron 20, only two members were assigned, POLI and PARP11. As both are involved in DNA damage repair[40,52] and appear to have a similar impact across different virus infections, our analysis might have identified a close functional relationship between these proteins in the context of virus infections. Furthermore, TRIM25 and ZC3HAV1 are the sole members of neuron 17. TRIM25 plays a crucial role in the early stages of the immune response to viral infections, primarily by activating the RIG-I pathway[53]. ZC3HAV1, also known as zinc-finger antiviral protein (ZAP), is a well-described restriction factor for several viruses through targeting viral RNAs[54–57]. TRIM25 has previously been shown to be a cofactor of ZC3HAV1[58–60], thus contextualizing the functional clustering of these two proteins into the same neuron. Additionally, the SOM identified ISGs with distinct activities. More specifically, LY6E, OAS3, RNA 2′,3′-cyclic phosphate and 5′-OH ligase (RTCB), and EIF2AK2 were placed in distinct positions in the SOM and are the sole members of neurons 2, 3, 6, and 13, respectively. RTCB, an atypical RNA ligase and integral core component of the tRNA-splicing ligase complex (tRNA-LC), influenced the infection dynamics of multiple viruses, including HSV-1, MeV, RVFV, SARS-CoV-2, VACV, and VSV (Fig. 3c). Importantly, the depletion of RTCB did not significantly affect cell growth within 72 h of observation (Supplementary Fig. 4b). These results prompted us to validate the activity of RTCB. To understand the molecular mechanisms underlying the proviral activity of RTCB, we employed quantitative proteome expression analysis. We pre-treated RTCB KO

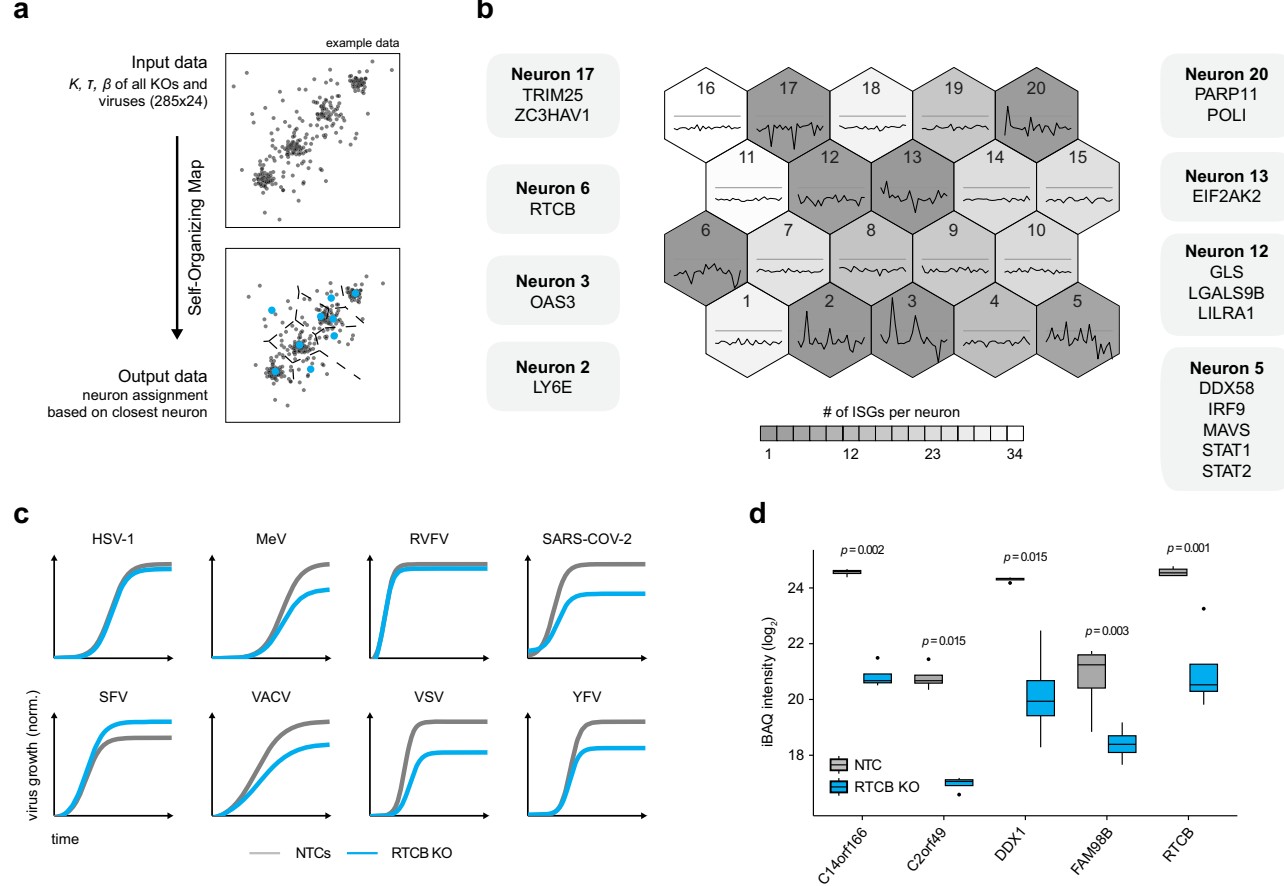

**Fig. 3 | Categorizing ISG specificity based on self-organizing map. a** Self-Organizing Maps (SOMs) project high-dimensional data onto a lower-dimensional grid of neurons. During training, the input data ($K$-, $\tau$-, and $\beta$-values across all eight viruses and 285 ISG KOs) are clustered to represent, as faithfully as possible, the similar localization of the data points in the multi-dimensional space into the grid of neurons. Each KO datapoint is then associated with the closest neuron. **b** SOM with a total of 20 neurons. The background color indicates the number of ISG KOs assigned to a particular neuron. The line plot within each neuron shows the codebook of that neuron, reflecting the weight vector derived from the input data. A high-weight vector value indicates that the input parameters strongly influence the response of the neuron, suggesting a pronounced effect of the corresponding ISG on viral infection kinetics. Neurons with pronounced effects are further detailed, displaying their members surrounding the map. **c** Virus-derived fluorophore signals normalized to the H2B-mRFP or -eGFP signal measured by live-cell imaging. The sigmoidal curves, calculated from the mean parameters of all replicates, compare RTCB KO (cyan) with the average of five non-targeting control (NTC) samples (gray) for each virus. **d** Box plots showing iBAQ intensities of tRNA ligase complex (tRNA-LC) subunits in RTCB KO cells as determined by proteomic analysis. Box plots show the median (center line), first and third quartiles (box limits), and whiskers extending up to 1.5× the interquartile range. A two-tailed, two-sample Welch's $t$-test was performed ($n = 4$ biologically independent samples), followed by a false discovery rate (FDR) correction of $p$-values.

and NTC cells with 500 U/mL IFNα for 16 h to induce an ISG response and then used liquid chromatography coupled with tandem mass spectrometry (LC-MS/MS) to analyze proteomic changes. Notably, this analysis revealed a significant downregulation of all subunits of the tRNA-LC (C14orf166, C2orf49, DDX1, FAM98B, and RTCB) in RTCB KO cells[61] (Fig. 3d, Supplementary Fig. 4c, Supplementary Data 3). Functional enrichment analysis based on CORUM terms[62] revealed effects on the tRNA-LC, as well as terms related to the Spliceosome, the pre-rRNA complex, ribosomal subunits, and exosomes (Supplementary Fig. 4d, Supplementary Data 3). We expressed V5-tagged RTCB and control proteins in A549 cells and analyzed α-V5-precipitated proteins using LC-MS/MS. Among other proteins, the protein–protein interactome identified all five components of the tRNA-LC as significant interactors of RTCB (Supplementary Fig. 4e, Supplementary Data 4). Enriched CORUM terms of the interactors showed associations to chaperone-related complexes like the CCT complex, the BBS-chaperonin complex, and the CALM1-FKBP38-BCL2 complex (Supplementary Fig. 4f, Supplementary Data 4). This combined analysis suggests that RTCB binds to and stabilizes the tRNA-LC and its associated functions, while potentially being

involved in other pathways, such as RNA processing and protein folding.

In conclusion, the SOM approach provides a comprehensive overview of the complex activity of ISGs in viral infections and offers a predictive tool for future research. ISGs clustering to similar neurons may share similar properties, as illustrated by the grouping of functionally related ISGs, such as STAT1, STAT2, IRF9, and PRRs, that stimulate IFN induction and TRIM25 and ZC3HAV1, which are functionally connected. Moreover, the SOM allowed the identification of ISGs with unique functions, such as RTCB.

**Analysis of ISG activities against SARS-CoV-2**

Regularly occurring viral epidemics and pandemics necessitate a deeper understanding of the functional role of host factors, which have the ability to control these viruses. To visualize the impact of all ISGs on SARS-CoV-2, we plotted $K$-value (max fluorophore expression value) and $\tau$-value (time to reach $K/2$) obtained from all tested KO cells (Fig. 4a, b). As expected[17,50], the depletion of LY6E prominently increased the $K$-value. Similarly, our data recapitulated STAT2 and IRF9 as antiviral factors. Several other ISG KOs demonstrated pronounced

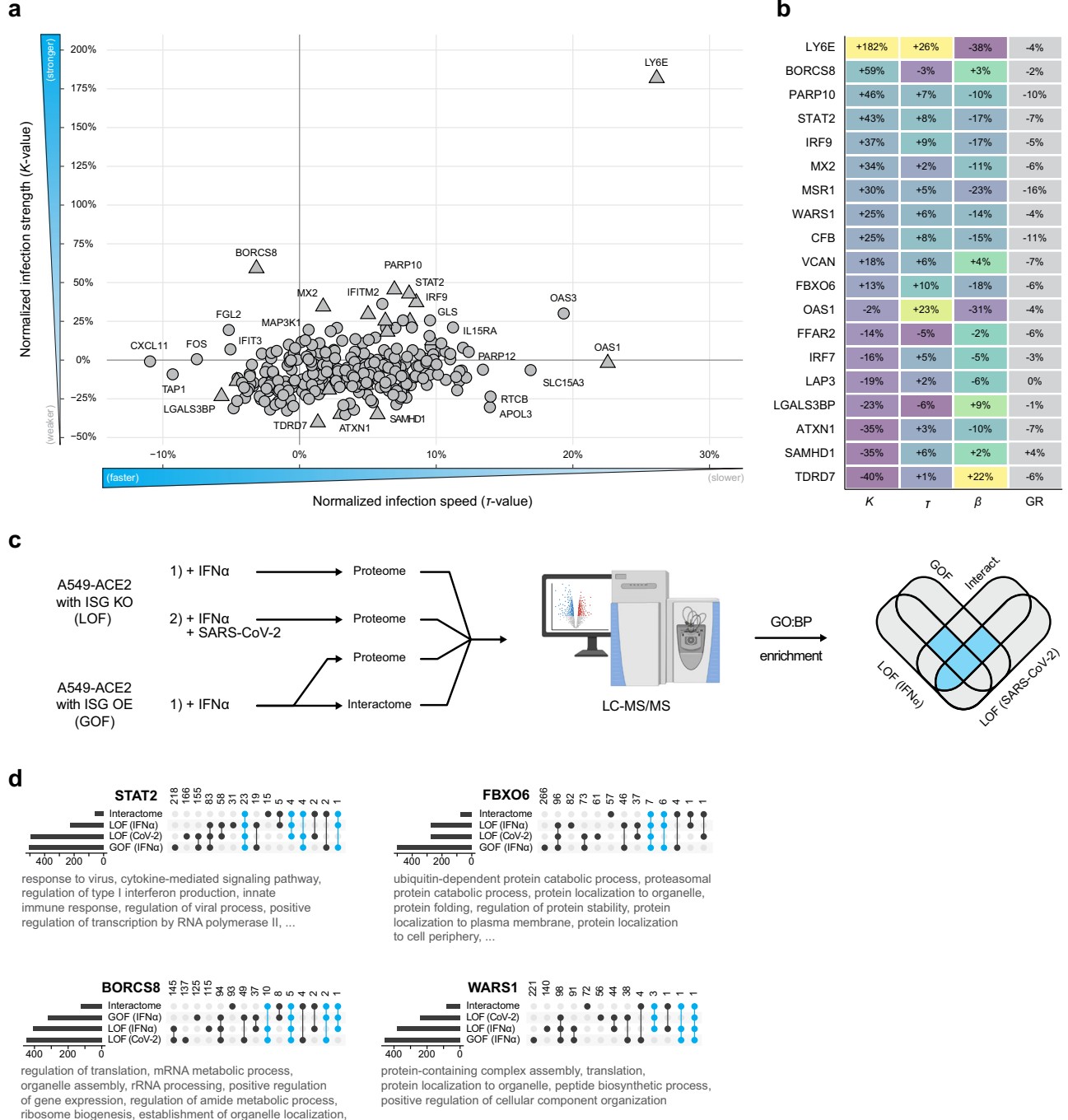

Fig. 4 | Analysis of ISG activities against SARS-CoV-2. a Scatter plot showing normalized infection strength (K-value) and speed (τ-value) for all ISG knockout (KO) cells infected with SARS-CoV-2. Triangles denote ISGs selected for follow-up experiments. b Table showing selected ISGs along with their respective K-, τ-, β-values, and growth rates; color gradients correspond to the column-specific minimum and maximum values. c Schematic of experimental design used for proteomic analysis of ISG KO and ISG over-expressing (OE) cells, loss-of-function (LOF), and gain-of-function (GOF), respectively. KO cells were pre-treated with IFNα (6 h, 6.25 U/mL), infected with SARS-CoV-2 or left uninfected for 24 h and subjected to proteome analysis. Cells overexpressing V5-tagged ISGs were pre-treated with 500 U/mL IFNα for 16 h. All samples were subjected to proteome and interactome analysis using α-V5 affinity purification. The resulting proteome data were

analyzed for enriched Gene Ontology Biological Process (GO:BP) terms. Intersections of terms identified in the interactome data and at least two other datasets (cyan intersects) were then used to describe the function of the ISG. Created in BioRender. Pichlmair, A. (2026) https://BioRender.com/3op6pws. d UpSet plots showing the intersection of enriched GO:BP terms derived from the significantly enriched proteins of the datasets depicted in c. The bar plots on the left of each UpSet plot show the total number of enriched GO:BP terms per dataset. The number of shared terms among datasets is shown at the top of each intersection. The terms present in at least two proteome datasets and the interactome dataset (cyan intersects) were selected and are shown below each plot. The terms were manually filtered for redundancy and generic terms.

antiviral activity against SARS-CoV-2, such as PARP10 and CFB. PARP proteins were shown to have antiviral activity against Alphaviruses[63–65], while deregulation of the complement system involving CFB was shown to play an important role in Dengue virus infection[66,67]. However, an antiviral activity against coronavirus has not been reported so far. Further unexpected findings were the antiviral activities of Tryptophanyl-tRNA Synthetase 1 (WARS1), which is essential to attach tryptophan to its cognate tRNA, and BORC subunit 8 (BORCS8, MEF2BNB), a protein involved in cellular trafficking and organelle biogenesis[68]. We also identified ISGs, which, when depleted, reduce virus infection dynamics. Among them was LGALS3BP, a protein reported as a negative regulator of the IFN response[45,69], which may explain the reduced SARS-CoV-2 infection observed upon its depletion. A similar reduction of SARS-CoV-2 propagation was observed by depletion of TDRD7, a protein implicated in stress granule formation, translational silencing, and autophagy[70]. TDRD7 is a restriction factor for VACV and VSV[71,72], which we could recapitulate in our screen (Fig. 1c). Unexpectedly, we could identify proviral activities for ATXN1, a protein involved in transcriptional repression and RNA splicing[73], and SAMHD1, best described for its antiviral activity against HIV-1 and other retroviruses by inhibiting viral DNA synthesis[69].

We reasoned that ISGs with pro- or antiviral functions are involved in pathways that are particularly important for SARS-CoV-2 replication. To identify such pathways, we performed orthogonal proteomic profiling of cells that lack or overexpress the 19 ISGs with the most prominent SARS-CoV-2 phenotypes (Fig. 4c). To determine the cellular interaction partners for the selected ISGs, we expressed the 19 candidates as V5-tagged proteins in A549 cells (Fig. 4b, c). Cells were stimulated with IFNα (500 U/mL, 16 h) to mimic the proteome composition of an antiviral response and subsequently used for affinity purification using α-V5 beads followed by LC-MS/MS analysis. We confirmed the successful identification of all bait proteins except for LY6E, which was not detected, and identified 10–40 interactors per bait (Supplementary Fig. 5a, b, Supplementary Data 5). Overlap with known interactors of SARS-CoV-2[74] with ISG interactors revealed 32 overlapping proteins in both datasets (Supplementary Fig. 5c), which could serve critical roles in mediating anti-SARS-CoV-2 activity. We further determined the differential protein expression pattern of A549 cells overexpressing the 19 ISGs (Fig. 4c), which reveals pathways that are regulated by the selected ISGs and mimics gain-of-function experiments (Supplementary Data 5). These data included 329 known SARS-CoV-2 interactors which could be relevant to mediate antiviral activity against SARS-CoV-2 (Supplementary Fig. 5d). We also characterized protein expression in cells depleted for the individual ISGs and pre-treated with IFNα (6.25 U/mL for 6 h) or pre-treated with IFNα followed by SARS-CoV-2 infection, respectively. We confirmed a significant depletion of eight ISGs, which were detectable by LC-MS/MS (Supplementary Fig. 5e). Among the 6,379 quantified proteins, we identified 50–150 proteins that were significantly regulated in each of the individual ISG KO cells (Supplementary Fig. 5f, Supplementary Data 6). As expected, depletion of STAT2 and IRF9 led to an impaired antiviral innate immune response pattern (e.g., reduced EIF2AK2, IFIT1, ISG15, OAS3, and STAT1) (Supplementary Fig. 5g). To concatenate these datasets and to filter for the dominant pathways regulated by the selected ISGs, we intersected GO:BP terms that were enriched in the interactome and at least two protein expression datasets (Fig. 4d, Supplementary Fig. 6a, Supplementary Data 7). This strategy successfully recapitulated the function of STAT2 as a key protein in the innate immune response, as indicated by the associated terms "type I interferon-mediated signaling" and "defense response to virus", among others. FBXO6 exhibited terms associated with ubiquitination, proteasomal degradation, and protein folding, aligning with its known function in endoplasmic reticulum-associated degradation[31,75]. Interestingly, the ubiquitin-dependent degradation of proteins associated with FBXO6 and its potential involvement in protein dynamics during

viral replication may explain the broad antiviral activity of FBXO6 (Fig. 1c, Fig. 4d). Similarly, the term "organelle localization" for BORCS8 matched its known function in lysosome localization and motility. Enrichment of other terms, such as RNA processing and translation regulation, suggests a multifaceted role of BORCS8 in controlling viral replication. Moreover, WARS1 was linked to RNA processing and translation, among other functions, which are also congruent with its role in protein synthesis.

Our analysis revealed a diverse range of cellular processes that contribute to the inhibition of SARS-CoV-2. Among the processes most significantly affecting SARS-CoV-2 growth are IFN signaling, protein stability and degradation, RNA processing, transcription and translation, and trafficking (Fig. 4d, Supplementary Fig. 6a). This suggests that targeting proteins or a combination of proteins involved in these processes would be valuable for perturbing virus infection. Moreover, we confirmed the antiviral activities of key ISGs such as LY6E, STAT2, and IRF9.

## Synergistic activity of ISGs

Since ISGs are co-expressed after exposure to type-I IFN, we tested whether the combined depletion of any of the 19 selected ISGs may potentiate their effect on SARS-CoV-2 propagation. We thus depleted the selected ISGs in all possible pairwise combinations and measured the progression of SARS-CoV-2-GFP replication by fluorescence live-cell imaging (Fig. 5a). We assumed a multiplicative relationship between the single KO effects and thus calculated the difference between the product of the two single effects to the experimentally evaluated effect of ISG co-depletion (Fig. 5b). The resulting interaction scores were then plotted in a heatmap using hierarchical clustering (Fig. 5c, Supplementary Data 8). Co-depletion of the majority of tested ISGs equaled the expectations considering the product of two individual KOs. The co-depletion of LY6E, BORCS8, and STAT2 with other candidates had particularly beneficial effects that exceeded the expected effects of the individual KOs. This was most prominent for the co-depletion of LY6E with either STAT2 or IRF9. These results indicate that concurrent activation of IFN signaling through STAT2 and inhibition of viral entry through LY6E exerts a synergistic antiviral effect against SARS-CoV-2. Similar synergisms were observed for the depletion of BORCS8 with STAT2 and IRF9, respectively. However, the co-depletion of some ISGs also resulted in lower-than-expected values, as indicated by negative interaction scores, suggesting that depleting both proteins did not confer any additional impact. This was particularly evident for the co-depletion of STAT2 and IRF9 and several other ISG combinations (ATXN1 with VCAN, ATXN1 with OAS1, OAS1 with WARS1, SAMHD1 with PARP10, and LY6E with BORCS8) (Fig. 5c). Plausible explanations for this phenotype may be that the respective activity of these ISGs may depend on each other or contribute to similar cellular functions. Indeed, STAT2 and IRF9 contribute to the same pathway (IFN signaling), which explains why their simultaneous depletion does not lead to a greater effect compared to single depletions. A similar relationship could be envisioned for, e.g., SAMHD1, which regulates the intracellular nucleotide pool, and PARP10, which requires nucleotides for poly-ADP ribosylation. Interestingly, the co-depletion of BORCS8 and LY6E did not lead to the expected increase in antiviral activity. This led us to hypothesize that BORCS8 and LY6E may operate in a similar pathway or inhibit the same stage of the viral life cycle. Since LY6E inhibits virus entry[17,50], we envisioned that BORCS8 may be similarly involved in SARS-CoV-2 infection. To test this, we employed GFP-encoding VSV particles that were pseudotyped with the SARS-CoV-2 Spike protein (VSV-S), thereby utilizing the SARS-CoV-2 entry route into host cells. Depletion of BORCS8 significantly enhanced infection by VSV-S but not of VSV particles bearing their natural glycoprotein (G) (Fig. 5d), supporting the involvement of BORCS8 in SARS-CoV-2 entry.

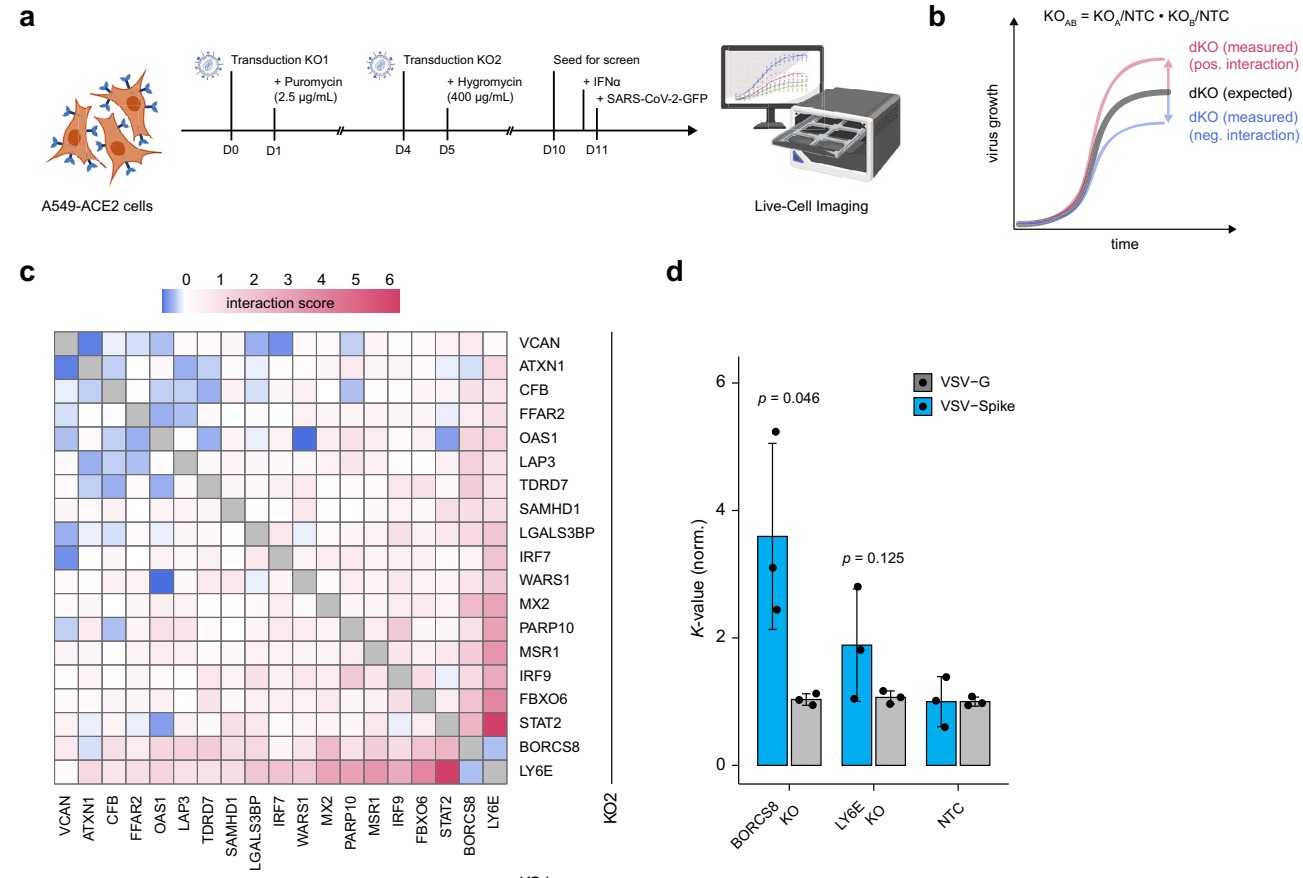

**Fig. 5 | Synergistic activity of ISGs. a** Diagram illustrating the co-depletion of ISGs and evaluation of their impact on SARS-CoV-2-GFP infection. ISG knockouts (KOs) were performed sequentially by transducing cells with CRISPR/Cas9 lentiviral vectors encoding puromycin or hygromycin resistance. After selection, the co-depleted KO cells were infected with SARS-CoV-2-GFP (MOI 3). Fluorescence images were acquired using live-cell fluorescence imaging every 4 h for 72 h. Created in BioRender. Pichlmair, A. (2026) https://BioRender.com/fgxwlao. **b** Schematic to evaluate interaction scores. For each replicate, $K$-values were normalized to the median value across all co-depleted KO cells. $K$-values of ISG KOs paired with an NTC were considered as single KO effects. We estimated the anticipated double KO (dKO) effect, by multiplying these individual effects. Subsequently, the interaction score was calculated by subtracting the estimated dKO effect from the observed dKO effect. A positive interaction score indicates a synergistic effect beyond the expected combined individual effects, while a negative score suggests an antagonistic interaction. **c** Interaction scores of all dKO combinations, displayed following hierarchical clustering. Redundant combinations represent duplicated values. Shown is the mean ($n = 4$ biologically independent samples). **d** GFP-expressing VSV-Spike pseudotyped virus infection compared to VSV-G in BORCS8 KO, LY6E KO, and NTC cells. All data are normalized to the corresponding NTC cells, as measured by live-cell fluorescence imaging. Peak signal intensity was observed at 24 h for VSV-G and 48 h for VSV-Spike. Data are presented as mean ± SD. One-sided, two-sample Welch's $t$-test ($n = 3$ biologically independent experiments).

Through this systematic analysis, we demonstrated that co-depletion of specific antiviral ISGs, such as LY6E with STAT2 or IRF9, prominently supports SARS-CoV-2 growth, indicating potent synergistic activity against SARS-CoV-2 when concomitantly disrupting viral entry and stimulating IFN signaling.

**BORCS8 impacts vesicle trafficking and endosomal dynamics**

We decided to further evaluate the role of BORCS8 on SARS-CoV-2 infection. The effect of depleting BORCS8 showed a prominent effect for SARS-CoV-2 in which the KO led to increased virus infection. We also observed moderate antiviral effects for HSV-1 and YFV (Fig. 1c, Supplementary Fig. 7a). Analysis of the genomic locus by Sanger sequencing confirmed correct targeting of BORCS8 (Supplementary Fig. 7b) and the depletion of BORCS8 did not affect growth rates as compared to NTC cells (Supplementary Fig. 7c). In line with the screening data, SARS-CoV-2 infected BORCS8 KO cells accumulated significantly more infectious virus particles in their supernatants for the parental strain and variant of concern (VOC) B.1.617.2 (Delta variant) of SARS-CoV-2, as compared to infected control cells (Fig. 6a). Similarly, BORCS8 depletion yielded significantly higher viral RNA levels in

accumulation of VOCs B.1.617.2 (Delta variant) and BA.1 (Omicron variant) as measured by RT-qPCR (Supplementary Fig. 7d). Compared to NTC cells, CRISPR/Cas9-mediated depletion of BORCS8 in Calu-3 cells also led to an increase in SARS-CoV-2 growth, which was comparable to the depletion of LY6E (Fig. 6b), suggesting that BORCS8 exerts antiviral activity across cell types. Expression of CRISPR/Cas9-resistant BORCS8 in BORCS8 KO cells reversed the enhanced viral replication, confirming the specificity of the KO approach (Fig. 6c).

The depletion of BORC subunits has been shown to result in a dysfunctional BORC complex, leading to impaired lysosomal motility, which can be characterized by the perinuclear localization of LAMP1-stained vesicles[68,76]. Indeed, Airyscan microscopy of LAMP1 revealed a pronounced perinuclear accumulation of lysosomes in BORCS8 KO cells, confirming functional targeting of the BORC complex (Fig. 6d). Lysosomes are known to be relevant for SARS-CoV-2 entry and egress. We hypothesized that a potential inhibitory effect of BORCS8 on endosomal uptake of SARS-CoV-2 can be alleviated by facilitating plasma membrane fusion of SARS-CoV-2 by expressing TMPRSS2 in BORCS8 KO cells. As expected, TMPRSS2 expression increased viral RNA levels in NTC cells, indicating enhanced plasma membrane entry

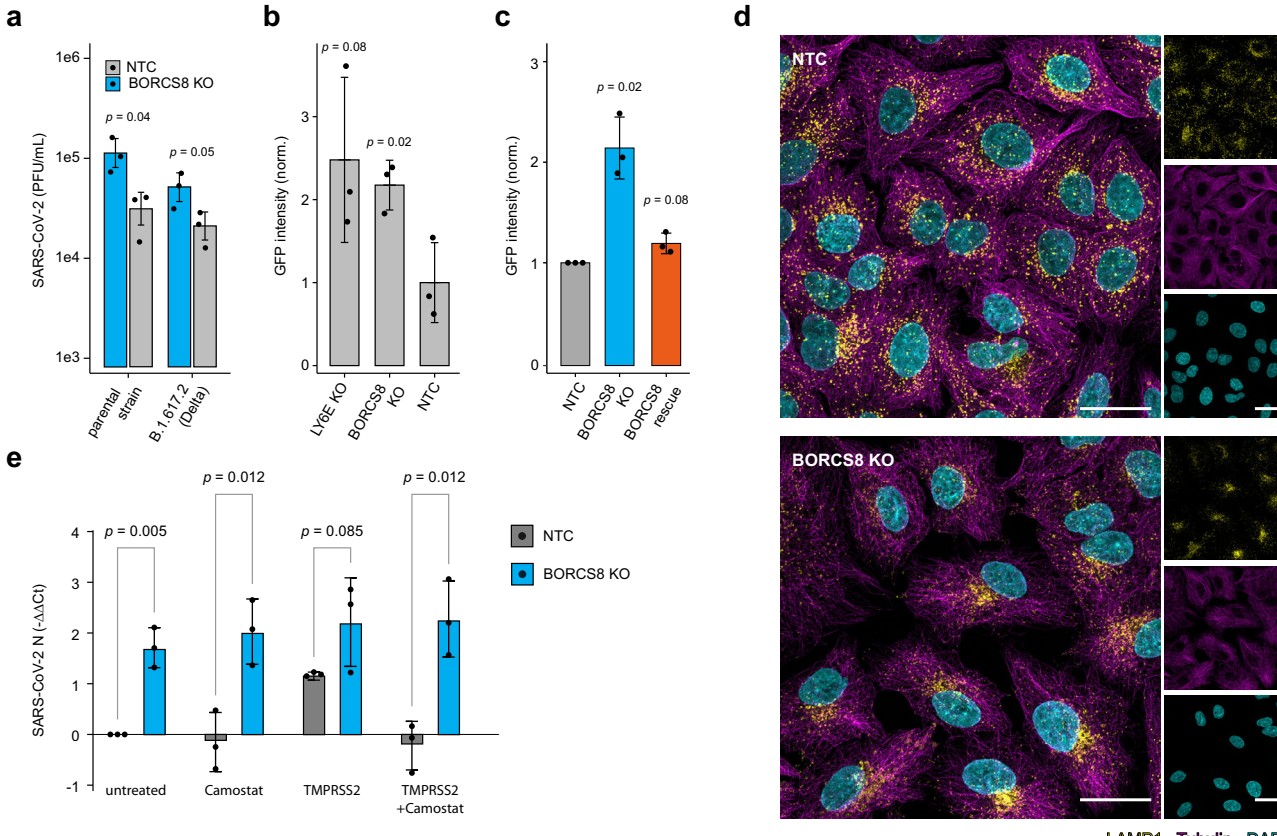

**Fig. 6 | BORCS8 impacts vesicle trafficking and endosomal dynamics. a** Plaque assay comparing SARS-CoV-2 accumulation in BORCS8 knockout (KO) and NTC supernatants infected with the parental strain and B.1.617.2 (Delta) variant of SARS-CoV-2 (MOI 3) at 24 h post-infection. Shown is the mean ± SD ($n = 3$ biologically independent experiments). One-sided, two-sample Welch's $t$-test. **b** Calu-3 cells were infected with SARS-CoV-2-GFP (MOI 3), and the GFP signal was measured by live-cell imaging. The figure shows the integrated intensity of the virus-expressed GFP signal normalized to the confluence of the cells at 60 h post-infection. Data are presented as mean ± SD ($n = 3$ biologically independent experiments). One-sided, two-sample Welch's $t$-test. **c** A549-ACE2 cells of the indicated genotype were infected with SARS-CoV-2-GFP (MOI 3) and imaged at 2 days post-infection. GFP intensity was normalized to NTC. Mean ± SD ($n = 3$ biologically independent experiments). One-sided, one-sample $t$-test. **d** Representative Airyscan images of NTC and BORCS8 KO cells stained for LAMP1, α-Tubulin, and DAPI. Maximum intensity projection. Scale bar: 25 μm. **e** Effect of TMPRSS2 expression and Camostat treatment on SARS-CoV-2 N RNA levels in BORCS8 KO and NTC A549-ACE2 cells at 24 h post-infection. Cells were treated with 10 μM Camostat for 6 h prior to the infection. SARS-CoV-2 N RNA was quantified by RT-qPCR and normalized to the mean of NTC samples. Bars show mean ± SD of −ΔΔCt values ($n = 3$ biologically independent experiments). One-way ANOVA, followed by Tukey's HSD post hoc test for multiple comparisons. SD standard deviation.

(Fig. 6e). The TMPRSS2 inhibitor camostat reversed this effect, confirming specificity. Notably, expression of TMPRSS2 or camostat treatment had no effect in BORCS8 KO cells, indicating that BORCS8 controls the endosomal uptake route of SARS-CoV-2.

Previously, ORF3a of SARS-CoV-2 was shown to associate with lysosomes to exploit lysosomal exocytosis for viral egress[76–78]. The fusion of lysosomes with the plasma membrane releases nascent virus particles into the extracellular space while simultaneously transferring lysosome-associated proteins onto the plasma membrane. Since BORCS8 depletion affected the lysosomal compartment (Fig. 6d), we investigated whether BORCS8 might influence the correct localization of ORF3a. As expected, exogenously expressed ORF3a localized to both the cytoplasm and the plasma membrane in control cells (Supplementary Fig. 8a). In BORCS8 KO cells, however, the localization of ORF3a was exclusively cytoplasmic, indicating defective ORF3a-trafficking due to BORCS8 depletion. A prominent role of BORCS8 in endosomal trafficking was also evident from proteome analysis of cells overexpressing or depleted for BORCS8 (Fig. 4d). Differentially expressed proteins and associated enrichment terms of IFNα pre-treated BORCS8 KO cells indicated perturbations of proteins involved in translation and in the BLOC1/2 complex, which is critical for endosomal cargo sorting, and the endosomal targeting complex (Fig. 7a).

Moreover, BORCS8 KO cells infected with SARS-CoV-2 revealed perturbation of the CCC and the CCC-Wash complex (Fig. 7b), which mediate endosomal sorting, protein trafficking, and actin polymerization on endosomal membranes[79]. BLOCS8 co-precipitated with the majority of BORC complex subunits (BORCS5, BORCS8, C17orf59, C10orf32, KXD1, and LOH12CR1) but also components of the related BLOC complex (BLOC1S1, BLOC1S2, and SNAPIN) (Fig. 7c). The associated enrichment terms further emphasize the role of BORCS8 in the BORC and BLOC complexes, and the v-ATPase-Ragulator complex, an essential component for lysosomal function (Fig. 7d). Additionally, the interactome of BORCS8 in the IFNα-treated condition enriched for the TRBP-containing complex, which is involved in RNA silencing, and the TNF/NF-κB signaling complex 6, which is integral for inflammatory responses (Fig. 7e).

Based on the involvement of BORCS8 in endosomal cargo sorting and given the increased entry of Spseudotyped VSV particles in BORCS8-depleted cells, we specifically focused on the role of BORCS8 in endocytosis. We treated cells with LysoTracker Red, a pH-sensitive dye that stains acidic vesicles in the cell, and performed live-cell imaging of BORCS8 KO and control cells. Notably, compared to controls, BORCS8-deficient cells showed increased levels of acidified vesicles, as indicated by the integrated intensity of LysoTracker (Fig. 7f).

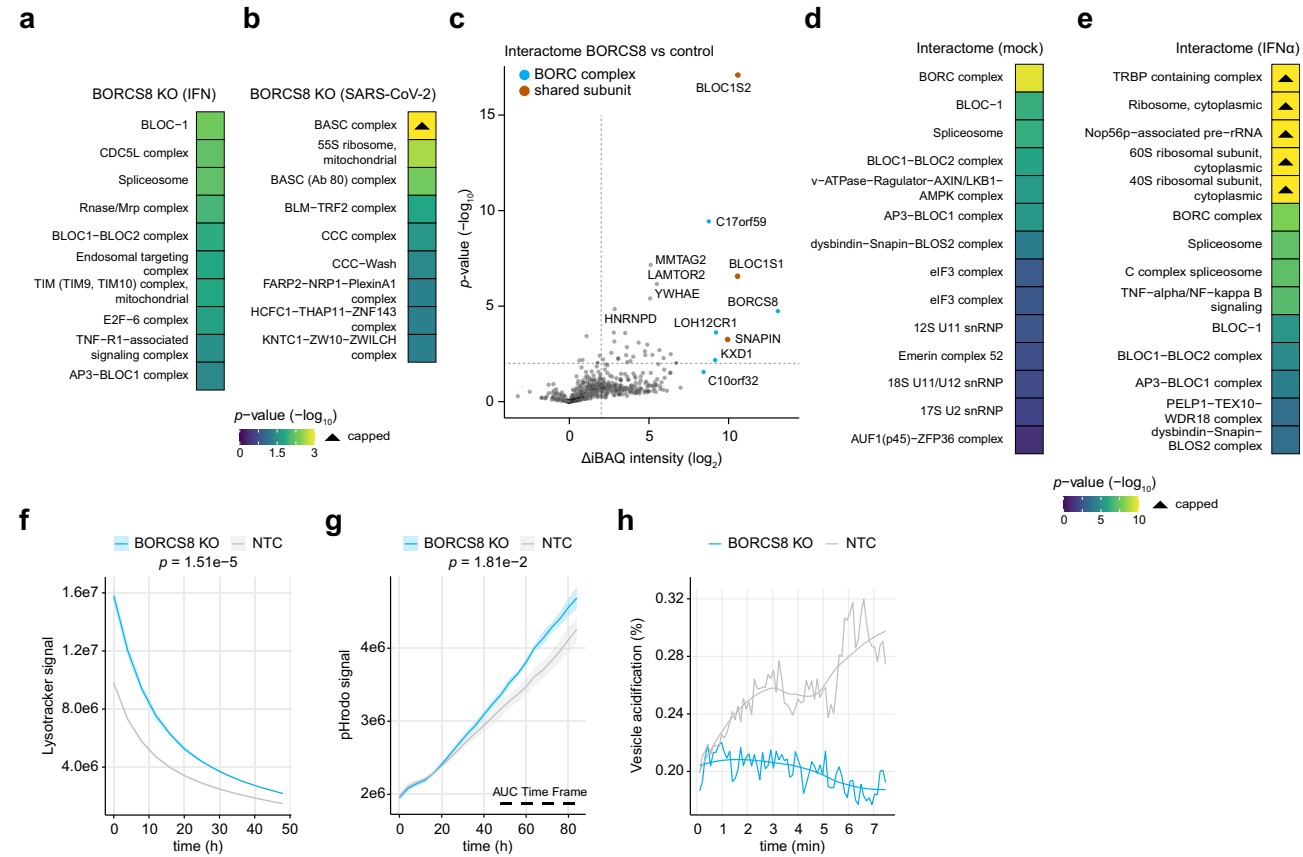

**Fig. 7 | Analysis of BORCS8 function during SARS-CoV-2 infection. a–e** A549-ACE2 cells were analyzed as shown in Fig. 4c. a-b and d-e. Gene set enrichment analysis based on the indicated proteomic dataset (4 biologically independent samples) using g:Profiler and a one-sided hypergeometric test. *p*-values were adjusted for multiple comparisons using the g:SCS algorithm. The most significant CORUM terms are shown. *p*-values exceeding the maximum range are capped and indicated with a triangle. **c** Volcano plot showing protein interactors detected by interactome analysis in A549 cells overexpressing BORCS8. Subunits of the BORC and BLOC complex are indicated in cyan and orange. Two-tailed, two-sample Welch's *t*-test (*n* = 3 biologically independent samples). Protein groups with an | log₂FC | ≥ 2 and a *p*-value ≤ 0.05 were considered significant and are indicated by a dotted line. **f** Time-course analysis of acidic vesicles in BORCS8 KO and NTC cells stained with LysoTracker Red. The mean ± SD is shown as line and ribbons (*n* = 4 biologically independent samples). The area under the curve (AUC) was calculated for each replicate and subjected to a two-tailed, two-sample Welch's *t*-test. **g** As f), but cells were treated with pHrodo Green. **h** Time-resolved colocalization of endocytosed cargo (pH-insensitive dye, Dextran-AF647) with endocytosed and acidified cargo (pH-sensitive dye, pHrodo Green). Confocal images were acquired every 6.5 s for a total of 7 min. The colocalization was carried out by Otsu's thresholding for each channel and time-point and calculating the number of double-positive pixels normalized to the number of Dextran-AF647-positive pixels (total endocytosed cargo). Individual values and the loess-smoothed trend over time are shown.

Reconstitution of KO cells with BORCS8 reversed the increased Lyso-Tracker signal to levels observed in control cells (Supplementary Fig. 8b). The increased acidification of endosomes could arise from malfunctions in either the maturation or fusion processes of endosomes and lysosomes, which are crucial for the degradation of endocytosed cargo and the recycling of cellular components. In line with this, we observed higher levels of vesicles labeled with pHrodo, a pH-sensitive dye that is internalized by endocytosis, allowing for the evaluation of early endosome acidification (Fig. 7g). In BORCS8-depleted cells, endocytosed cargo accumulated to higher levels, indicating retention of endocytosed cargo in the process of endosome maturation. Interestingly, when evaluating time-resolved colocalization of a pHrodo and a pH-insensitive dye Dextran Alexa Fluor 647, representative of endosome acidification and endocytosis activity, respectively, we could observe a defect in the acidification of incoming cargo in BORCS8 KO cells (Fig. 7h, Supplementary Fig. 8c). However, acidified vesicles and the contained cargo appear to not progress into more mature endo-lysosomes, which aligns with a role of BORCS8 in endocytosis and lysosomal function. Collectively, depletion of BORCS8 alters endo-lysosomal dynamics and enhances endocytic uptake, suggesting that BORCS8 constrains SARS-CoV-2 infection by maintaining proper vesicle processing and restricting entry via the endocytic pathway.

## Discussion

The innate immune system has evolved an array of ISGs with diverse functions to impair virus propagation. A comprehensive understanding of the IFN response and the activity of individual ISGs is crucial for comprehending the complex interplay between the host and viruses. Noteworthy advancements were made through pioneering work employing high-throughput overexpression screens[13,14]. While these studies were instrumental in identifying the activity of a large subset of ISGs, overexpression only partially reflects their activity since individual protein expression may not lead to the formation of functional complexes. To overcome this limitation and gain a deeper understanding of the complex interactions between ISGs and viruses, the primary objective of this study was to elucidate these relationships and propose effective concepts for targeting virus replication by utilizing an arrayed loss-of-function screening approach.

Our analysis revealed diverse anti- and proviral activities of 285 ISGs against eight viruses in a time-resolved manner (Fig. 1c). Confidence in the data stems from the identification of well-established

antiviral activities of certain ISGs and the selective activities of some ISGs against individual viruses or subsets of viruses. Although this dataset indicates effects of individual ISGs against diverse viruses, the limitations of the screening system may not directly be comparable to similar quantitative biological effects of gene functions in other systems. Importantly, we identified a substantial number of ISGs with yet unreported antiviral and proviral effects. Differential virus growth can arise from direct ISG anti- or proviral functions or be mediated through ISG-regulated cellular pathways or transcriptional processes. The transcription factors STAT1, STAT2 and IRF9, and the PRR DDX58 (RIG-I), for instance, are not directly antiviral per se, but induce antiviral proteins (ISGs) that restrict viral growth. ISGs are classically understood as antiviral effectors; the surprisingly large number of ISGs with pro-viral activity indicates a broader involvement of ISGs in cellular processes, some of which may be co-opted by viruses to enhance replication or evade immune responses. While we could recapitulate the antiviral activities of many previously studied ISGs, the presented resource identified multiple proteins that have not been reported to have anti- and proviral activity. Multiple ISGs exhibited antiviral activity against distinct virus classes or individual viruses, showcasing a wide array of antiviral mechanisms and specificities employed by the innate immune system (Fig. 2). Surprisingly, several ISGs, such as POLI, LGALS9B, and ZC3HAV1, are antiviral for some viruses but are proviral for others (Fig. 1c). The scope of this study enabled the identification of inter-viral relationships and revealed functional linkages across distinct ISGs (Fig. 3b). The functional similarity of yet under-explored ISGs may enable to draw mechanistic relationships between proteins, e.g., PARP11 and POLI, and point towards shared cellular pathways involved in their antiviral functions.

Importantly, while $K$- and $\tau$-values enable the identification of virus-modulating ISGs and the classification of broad versus virus-specific trends, these parameters should not be interpreted as absolute or directly comparable measures of antiviral potency across ISGs. To accommodate differences in replication kinetics, cytopathogenicity, and reporter signal intensity, infections were performed at virus-specific MOIs. Consequently, all quantitative analyses and z-score normalizations were performed within each virus dataset, and comparisons across viruses are qualitative rather than quantitative. Moreover, differences in knockout efficiency, reporter signal dynamics, and infection conditions introduce unavoidable heterogeneity that limits quantitative ranking of ISG strength.

While the consistency of trends observed across replicates and the robust impact on viral infections support the reliability and relevance of our findings, a limitation of any screen of this size is the potential variation in KO efficiency. However, the identification of previously reported ISG activities and the successful confirmation of gene depletions for tested candidates at transcript and protein levels (Supplementary Fig. 1d, Supplementary Fig. 5g) indicate overall effective depletion of the targeted ISGs. While the data presented here highlights the anti- and proviral activities of ISGs, it is important to note that the antiviral activity of individual ISGs may vary between cell types due to differences in the expression levels of the ISGs themselves or their co-factors. ISGs with incomplete depletion may appear neutral or weakly acting in this dataset despite biologically relevant functions, whereas ISGs with efficient depletion and non-redundant roles are preferentially detected. Additional variability that cannot be excluded comes from potential effects of ISGs on cell growth and viability. We excluded ~5% of the tested ISGs due to negative effects on cell growth under steady state conditions. Some ISGs, such as APOBEC3A[80,81], are known to affect cell viability in an infection-dependent context. In our loss-of-function approach, we did not observe an apparent ISG-dependent influence on cell viability in infection settings, as judged by the signal of fluorescently labeled H2B proteins. A remaining limitation of this screening approach is the impact of ISG depletion on cell proliferation during an antiviral or inflammatory immune response

conditions. In infected conditions our screening data does not allow to delineate the impact of ISG depletion on cell growth from cytopathic effects caused by viral infection. Our study identified RTCB as a key factor for the replication of multiple viruses (Fig. 3b). RTCB, a core component of the tRNA-LC, is associated with XBP1 mRNA splicing during the unfolded protein response[61,82,83]. Recent studies showed that RTCB depletion impairs Influenza A virus propagation[84]. Interestingly, RTCB has also been shown to bind viral RNA of SARS-CoV-2 to modulate infection progression[85,86]. Unbiased proteomics analysis revealed that RTCB depletion results in the downregulation of all tRNA-LC subunits (Fig. 3d), underscoring the critical role of RTCB in maintaining the stability of this complex. The broad proviral effect of RTCB, observed in our screen, highlights its potential as a target for future antiviral therapies and encourages further studies to dissect its precise role during viral infections.

Co-depletion experiments revealed both antagonistic and profound synergisms between individual ISGs (Fig. 5c). Proteomics profiling and additional functional experiments showed that ISGs inhibiting SARS-CoV-2 entry (e.g., LY6E, BORCS8) and ISGs regulating the IFN response (e.g., STAT2, IRF9) were highly effective in reducing SARS-CoV-2 replication. A dual-targeting strategy incorporating entry inhibitors and IFN stimulation could optimize treatment regimens. Indeed, recent clinical data on hepatitis D virus treatment showed that combining an entry inhibitor (myrcludex B) with type-I IFN treatment exhibited exceptional antiviral activity as compared to single treatments, which were ineffective in conferring long-term protection[87,88].

We focused on SARS-CoV-2 to identify restriction factors that limit infection. Notably, alongside the well-documented restriction factor LY6E, BORCS8 emerged as a novel modulator of the life cycle of SARS-CoV-2 (Figs. 4a, b, 6a-c). BORCS8 is a member of the BORC complex, a multiprotein complex that plays a critical role in the positioning, distribution, and motility of lysosomes as well as the fusion of lysosomes with autophagosomes[89–91]. The proper positioning and motility of lysosomes are essential for multiple cellular processes, including endocytosis, phagocytosis, and autophagy, as well as the biogenesis and function of lysosome-related organelles, which play a crucial role in the life cycle of many viruses, including SARS-CoV-2. In cell models lacking TMPRSS2 expression (e.g., A549 cells), the infection of SARS-CoV-2 requires progressive acidification of endosomes for viral genome release into the cytoplasm[92]. In BORCS8-depleted cells, we observed a defect of endocytosed cargo acidification (Fig. 7f-h). BORCS8 may thus regulate the transport of incoming cargo towards degradative lysosomes, thereby potentially limiting the efficiency of SARS-CoV-2 infection. Notably, BORCS8 was also active against HSV-1 and YFV (Supplementary Fig. 7a), which similarly rely on endosomal entry routes. SARS-CoV-2 exploits lysosomal exocytosis for virus egress[93], which has been reported to involve BORC-ARL8b complex-mediated anterograde transport of lysosomes to the plasma membrane[76]. It has recently been demonstrated that the SARS-CoV-2 protein ORF3a hijacks the lysosomal exocytosis pathway by recruiting BORC-ARL8 to lysosomes and facilitating viral egress[76,94]. The lack of plasma membrane-associated ORF3a in BORCS8 KO cells indicates a malfunction of the lysosomal exocytosis route (Supplementary Fig. 8a). This leads us to hypothesize a dual role for BORCS8 and the BORC complex in both virus entry and previously reported viral egress. Collectively, our data indicate that BORCS8 plays an important role in regulating endo-lysosomal functions, which are particularly relevant for virus entry. It remains possible that the contribution of BORCS8 to viral egress is context-dependent or becomes more relevant in later stages of infection.

In conclusion, the extensive resource provided herein, detailing ISGs and their virus-modulating activities, serves as a foundation for advancing our understanding of the innate immune system and offers a rational platform to explore the interplay of ISGs with novel pathogens, thereby aiding in the optimization of treatment strategies. The

co-depletion experiments offered yet undescribed insights into this complex interplay of ISGs with SARS-CoV-2. Together with complementary proteomics datasets, this resource enhances our understanding of the innate immune response organization, providing a rationalized blueprint for characterizing novel pathogens and optimizing treatment strategies.

## Methods

Research conducted within this study complies with all relevant ethical regulations and was approved by the TUM institutional review board.

### Cell lines and culture conditions

A549 (ATCC, CCL-185), A549-ACE2[74], HEK293T (ATCC, CRL-11268), Vero E6 (ATCC, CRL-1586), HFF (ATCC, SCRC-1041), SK-N-SH (ATCC, HTB-11) and Calu-3 cells (ATCC, HTB-55) (kindly provided by Stephan Pöhlmann, Deutsches Primatenzentrum, Münster, Germany, and Stephan Ludwig, University of Münster, Germany) were maintained in a humidified incubator at 37 °C and 5% $CO_2$ atmosphere as described previously. Cells were cultured in Dulbecco's Modified Eagle Medium (DMEM;; high glucose, pyruvate; Gibco Fisher Scientific), supplemented with 10% fetal bovine serum (FBS) and 1% penicillin-streptomycin (Sigma-Aldrich, P4333). All cell lines were tested and confirmed to be mycoplasma-free. Work with human material was approved by the TUM Institutional Review Board (Nr.: 2026-143-W-KK).

### Generation of lentiviral particles

Lentiviral particles were produced using HEK293T cells, employing a second-generation lentivirus system as described previously[95], by co-transfecting the transfer plasmid together with pCMV-Gag-pol and pMD2-VSV-G at a ratio of 4:2:1. Polyethyleneimine (Polysciences, 24765) was used for transfection at a 3:1 ratio relative to the total DNA. The medium was exchanged 6–12 h post-transfection. After 48 h, lentiviral supernatants were harvested, passed through a 0.45 μm filter, aliquoted, and immediately snap-frozen in liquid nitrogen for long-term storage.

### Generation of reporter cell lines

A549 cells were engineered to express either H2B-mRFP or H2B-eGFP using a two-step lentiviral transduction approach. First, A549 cells were transduced with lentiviral particles bearing either pHIV-H2B-mRFP or pHIV-H2B-eGFP transfer plasmids. pHIV-H2BmRFP was a gift from Bryan Welm & Zena Werb (Addgene plasmid #18982; http://n2t.net/addgene:18982; RRID:Addgene_18982)[96], and pHIV-H2B-eGFP was a gift from Maria Pia Cosma (Addgene plasmid #91776; http://n2t.net/addgene:91776; RRID:Addgene_91776)[97]. Subsequently, cells exhibiting the highest mRFP or eGFP fluorescence were isolated via fluorescence-activated cell sorting (FACS). For experiments involving SARS-CoV-2-GFP infections, A549-H2B-mRFP cells were further transduced to introduce the human *ACE2* gene under zeocin (Thermo Fisher, R25001) selection to yield A549-ACE2-H2B-mRFP cells, as described previously[74].

### Selection of ISGs targeted for knockout

To select ISGs in an unbiased manner, we used the data from the Interferome database[24], which contains microarray data of the expression patterns of 18,116 genes in response to type-I IFNs in different model systems. We analyzed the expression data to identify genes with substantial functional relevance in the IFN response. Specifically, we considered genes that showed significant changes in expression ($p$-value < 0.05 and $|\log_2FC| \geq 2$) in a minimum of 50% of experiments for upregulated genes and at least 10% for downregulated genes. Such changes had to be evident in at least three datasets. The final list included 300 ISGs, of which 77% were shown to be upregulated, and 23% were shown to be downregulated in response to type-I

IFN treatment. Alongside the chosen ISGs, positive controls STAT1, MAVS, BST2, EIF2AK2, and RSAD2 were included.

### Guide sequence design and cloning

Guide sequences for each target gene were designed using the Synthego guide design tool (https://design.synthego.com) or ChopChop[98]. Additionally, 15 NTC sequences were sourced from the GeCKO v2.0 library[99] and pooled in the same way as the other constructs. The lentiCRISPRv2 plasmids bearing the synthesized sgRNA templates were cloned as described previously[95]. lentiCRISPR v2 was a gift from Feng Zhang (Addgene #52961; http://n2t.net/addgene:52961; RRID:Addgene_52961)[99]. In brief, guide template oligonucleotides were annealed, pooled, and then ligated into the BsmBI-linearized lentiCRISPRv2 plasmid. The equal distribution of the three plasmids per target was confirmed by RT-qPCR (Supplementary Fig. 1c). Lentivirus particles for these constructs were produced as described above.

### Generation and handling of knockout cell lines

A549-H2B-mRFP, A549-H2B-eGFP, or A549-ACE2-H2B-mRFP cells were plated one day prior to transduction. For the transduction process, lentiCRISPRv2-based lentiviruses were added at a multiplicity of infection (MOI) of 5. After 24 h, cells underwent puromycin selection at 2.5 μg/mL for 7 days. After this, cell counts were equalized by counting the mRFP- or eGFP-stained nuclei for each KO using the IncuCyte S3 live-cell imaging system. Based on these counts, cell numbers were adjusted, and equal amounts were seeded onto 96-well plates.

### Evaluation of cell growth

The impact of gene targeting on the cell growth of the individual populations was evaluated based on the accumulation of the H2B-eGFP signal over time in mock-infected cells in five biological replicates. For these cells, we assessed the cell growth as measured by the accumulation of the H2B-eGFP signal for each time point. We fitted the cell growth kinetics using Eq. (1).

$$cell\,growth(t) = a \cdot e^{b \cdot t} \tag{1}$$

Where $a$ quantifies the green integrated intensity of the continuously expressed H2B-eGFP fusion protein at t = 0 h, representing the initial number of cells, and $b$ represents the growth rate. We then calculated the adjusted $r$-squared value for the fit and flagged samples that had 1) a bad fit, defined by $r$-adjusted <0.9, indicating kinetics deviating from the expected exponential growth due to non-growing or dying cells, or 2) an initial value of $a < 0.7e6$. KOs with three or more replicates flagged by either of the criteria were excluded from further analyses. We then normalized the growth rate to the mean growth rate of five NTC samples within each plate and averaged these normalized values to obtain the final representative growth rate for each KO.

### Virus stock preparations

HSV-1-Lox-mCherry (17+ strain)[100] was a kind gift of Beate Sodeik (Hanover Medical School, Germany). Recombinant MeV-eGFP (EdTag strain) was a kind gift of Dr. Martin Kächele (Technical University of Munich, Germany) and was produced by infecting VeroE6 cells (MOI 0.0001) for 2 days. The supernatants were harvested and spun at 1000 $g$ for 10 min. Aliquots were stored at -80 °C. RVFV-Katushka (ZH548 strain)[101] was a kind gift of Friedemann Weber (Justus Liebig University, Giessen, Germany). SARS-CoV-2 MUC-IMB-1[102] (denoted as SARS-CoV-2 or parental strain), SARS-CoV-2 B.1.617.2 (Delta strain)[103] and SARS-CoV-2 BA.1 (Omicron strain)[104] were produced as described previously[103]. SARS-CoV-2-GFP was a kind gift of Volker Thiel (University of Bern, Switzerland) and was produced as described previously[74]. SARS-CoV-2 Spike-pseudotyped VSV-GFP (VSV-S) was a kind gift of Janine Kimpel (Innsbruck Medical University, Austria) and was produced as previously described[95]. All experiments with SARS-

CoV-2 were performed in BSL3 laboratories under the approval of the Government of Upper Bavaria, Germany (AZ: 55.1GT-8791.GT_2-365-10 and 55.1GT-8791.GT_2-365-20). SFV-2SG-mCherry (SFV6 strain) was generated as described previously[105] and a kind gift from Andres Merits (University of Tartu, Estonia). VACV-GFP (WR strain, V300)[106] was a kind gift from Dr. Joachim Bugert (Bundeswehr Institute of Microbiology, Munich, Germany) and was produced by infecting Vero E6 cells (MOI 0.001) for 3 days. Infected cells were scraped, sonicated, and resuspended in OPTI-MEM (Gibco Fisher Scientific, 31985070) before storage at −80 °C. VSV-GFP (Indiana strain) was produced as described previously[9]. YFV-Venus (17D strain) was a kind gift from Simon Rothenfusser (Division of Clinical Pharmacology, Ludwig Maximilian University of Munich, Germany). If not otherwise specified, the virus stocks were produced as follows: VeroE6 cells were infected (MOI 0.01) for 3 days or until cytopathic effects were visible in approximately 50% of the cells. The supernatants were harvested and spun at 1,000 g for 10 min. Aliquots were then stored at -80 °C. All virus stock titers were determined by plaque assay.

### IFNα and inhibitor pre-treatment and reporter virus infection

A549-H2B-mRFP or A549-H2B-eGFP cells, with or without exogenous expression of ACE2, were pre-treated with virus-specific doses of recombinant IFNα b/d (IFNα)[107], which was a kind gift from Peter Stäheli. To assess the IFN response in A549 cells and to determine the IFN-sensitivity of the selected viruses, we experimentally determined the half-maximal effective concentration (ED50) of IFNα for each virus while also ensuring a comparable ISG induction across all viruses (Supplementary Fig. 1a). The determined doses for each virus were: HSV-1 at 50 U/mL, MeV at 50 U/mL, RVFV at 3 U/mL, SARS-CoV-2 at 6.25 U/mL, SFV at 2 U/mL, VACV at 50 U/mL, YFV at 1 U/mL, and VSV at 0.5 U/mL. The cells were then infected with the following MOIs: HSV-1 at 0.5, MeV at 0.01, RVFV at 1, SARS-CoV-2 at 3, SFV at 3, VACV at 0.05, VSV at 0.05, and YFV at 1.

For infections with pseudotyped VSV, A549-ACE2-H2B-mRFP cells, with ISG depletion or NTC controls, were seeded and 24 h later infected with either VSV-G at MOI 0.05 or pseudotyped VSV-Spike at MOI 3 in technical duplicates. Post-infection fluorescence images were acquired every 2 h for 72 h in total using the IncuCyte S3 system.

To inhibit TMPRSS2, cells were treated with 10 μM camostat mesylate (Santa Cruz, sc-203867), or DMSO for 6 h prior to infection with SARS-CoV-2 at MOI 1.

### Generation of overexpression constructs

To generate doxycycline-inducible overexpression constructs, full-length cDNA sequences encoding the target genes were synthesized by Twist Bioscience in the pTwist ENTR Kozak vector backbone. Target transcript variants and the used sequences as well as the codon-optimized sequence of BORCS8 are listed in Supplementary Data 1. Entry clones underwent Gateway LR recombination to transfer the target ORFs into the doxycycline-inducible and V5-tagged lentiviral destination vector pLIX403. pLIX_403 was a gift from David Root (Addgene plasmid #41395; http://n2t.net/addgene:41395; RRID:Addgene_41395). All final expression constructs were validated by Sanger sequencing prior to downstream applications.

### ISG and TMPRSS2 expression by stable transduction

Where indicated, A549 cells or A549-ACE2-H2B-mRFP BORCS8 KO cells were transduced with lentiviral particles that allow doxycycline-inducible expression of ISGs. To restore BORCS8 expression in BORCS8 KO cells, we transduced cells with lentiviral particles encoding a CRISPR/Cas9-insensitive, codon-optimized BORCS8 expression construct. After 24 h, cells were subjected to selection with hygromycin at 400 μg/mL for 7 days. Following selection, ISGs were expressed by treating cells with 1 μg/mL doxycycline (Sigma-Aldrich, D5207) for 24 h. In experiments involving doxycycline-inducible

expression constructs, both ISG-expressing cells and corresponding NTC cells were treated with doxycycline.

A549-ACE2-H2B-mRFP cells stably expressing TMPRSS2 were generated by two consecutive rounds of retroviral transduction and described earlier[95].

### Image acquisition and modeling of viral infection kinetics

We employed a quantitative analysis approach to understand the effects of specific ISG KOs on the progression of viral infections. Epi-fluorescence images were acquired every 2–3 h for 48–72 h using the IncuCyte S3 live-cell imaging system (2020C Rev1) with an S3/SX1 G/R optical module. Image analysis was performed using the IncuCyte S3 software (2020C Rev1). As indicated above, green and red object masks were generated by thresholding for fluorescent signals above roughly three times the fluorescence level of background noise. To quantify the virus infection dynamics, the integrated signal of green/red-positive objects per image, representing infected cells, was normalized to the integrated signal of red/green-positive objects per image, representing the cell number. Before fitting, we applied signal preprocessing functions to eliminate noise in the data. First, we observed a transient peak at t = 0 h from some viruses, which decayed to the baseline value before t = 9 h. This transient noise is an artifact due to the low signal-to-noise ratio, especially pronounced at early time points. In order to properly model the signal, we replaced the data points from 0–9 h to align with the data point at t = 9 h. Second, in rare cases, there were signal drops due to misfocused images. We mitigated this by applying a median filter (kernel size = 3) followed by a savgol filter (window length = 7, polyorder = 4) to remove any spikes or drops in the data. The resulting signal could then be modeled as a sigmoid curve and fitted according to Eq. (2):

$$f(t) = c + \frac{K}{1 + e^{-\beta(t-\tau)}} \quad (2)$$

Using this equation, each resulting sigmoid curve is characterized by a lower horizontal asymptote or baseline value $c$ and a maximum horizontal asymptote $c + K$, where $K$ represents the peak value that the curve reaches minus the baseline $c$. Unless otherwise stated, ISGs were designated as pro- or antiviral based primarily on $K$-values, reflecting changes in the overall infection-associated reporter signal. $\tau$ is the inflection point of the sigmoid curve and is derived by Eq. (3). It describes the point where the curve reaches half its peak value.

$$f(t = \tau) = c + \frac{K}{2} \quad (3)$$

$\beta$ quantifies the slope of the curve at t = $\tau$ and represents the progression rate of the virus infection. We then estimated the parameters $K$, $\tau$, and $\beta$ using the curve_fit function of the SciPy package[108]. Following the same normalization method as in the cell growth kinetics, we normalized the parameters to the mean of five NTCs within their corresponding plate and averaged these normalized values to obtain the representative parameters for each KO, using Eq. (4).

$$Avg. \, Normalize \, parameter \, KO_i = \frac{1}{n} \sum_{plate=1}^{n} \frac{parameter \, KO_i^{plate}}{NTCs^{plate}} \quad (4)$$

### Self-organizing map

We employed a self-organizing map (SOM) to reduce the dimensionality of our screen data and to reveal latent patterns shared across different KOs[109,110]. As input data, we used all z-scored parameters ($K$-, $\tau$-, and $\beta$-values of all KOs and viruses) and ran the SOM utilizing the kohonen R package with default settings, except the

learning rate, which was set to decline from $\alpha = 0.25$ to 0.01. We utilized a hexagonal topology with 20 neurons and 1,000 iterations for the SOM to learn the underlying data structure.

### RT-qPCR analysis and plaque assay

Cell lysis and RNA extraction were performed using the NucleoSpin RNA Plus kit (Macherey-Nagel, 740984), following the manufacturer's guidelines. Subsequently, total RNA was subjected to reverse transcription employing the PrimeScript RT kit with a gDNA eraser step (Takara, RR047B). Quantitative assessment of transcript levels was conducted using PowerUp SYBR Green Master Mix (Applied Biosystems, A25741). All primers used are listed in Supplementary Data 1. To validate KO efficiency, A549-H2B-mRFP KO cells were treated with 50 U/mL IFNα for 6 h and expression of a subset of ISGs was analyzed by RT-qPCR.

Plaque assays were conducted in confluent monolayers of Vero E6 cells, which were exposed to a fivefold serial dilution of viral supernatants and incubated for 60 min at 37 °C. Following incubation, the inoculum was aspirated and substituted with serum-depleted Minimum Essential Medium (MEM) (Gibco, Life Technologies, 11095080), supplemented with 0.5% carboxymethylcellulose (Sigma-Aldrich, 21904). At 48–72 h post-infection, the cells were fixed for 20 min using formaldehyde at ambient temperature. The fixed cells were then washed with Phosphate-Buffered Saline (PBS) (Gibco; Life Technologies, 14040133). Staining was performed using a solution of water, 1% crystal violet, and 10% ethanol for a period of 20 min. After a final rinse with PBS, plaques were counted, and the viral titer was determined.

### Sample preparation of proteome samples for mass spectrometry

We performed a mass spectrometry analysis to understand the impact of the selected KOs on the cellular proteome. Sample preparation was performed as described previously[74,111]. In brief, 2e5 KO or NTC cells were seeded on 6-well plates the day before the infection. The cells were then pre-treated with 6.25 U/mL IFNα 6 h before infection with SARS-CoV-2 (MOI 2) for 24 h. The samples were then harvested by aspirating the medium and washing the cells with PBS, followed by temporarily freezing at −80 °C until sample preparation. After thawing on ice, the cells were lysed with 200 µL sodium deoxycholate (SDC) (Sigma-Aldrich, 30968) lysis buffer and boiled for 5 min at 95 °C, shaking at 1400 rpm. Samples were then sonicated on a Bioruptor (10 min, 4 °C, 30 s on, 30 s off, high setting; Bioruptor, Diagenode), and the total protein concentration was normalized to 50 µg in a fixed volume using the Pierce BCA Protein assay according to the manufacturer's protocol (Thermo Fisher, 23225). The samples were then reduced and alkylated at 45 °C for 5 min with 10 mM TCEP and 40 mM CAA in the dark. Samples were pre-digested with Lys-C (1:100, Wako Chemicals, 4548995075888) for 1 h at 37 °C before trypsin (1:1000, Sigma-Aldrich) was added and incubated overnight at 37 °C. After that, the samples were diluted to a final volume of 200 µL using 1% trifluoroacetic acid (TFA) in isopropanol. The samples were then loaded onto SDB-RPS StageTips (Empore) and washed with 200 µL of 1% TFA in isopropanol, followed by 200 µL of 0.2% TFA in 2% acetonitrile (ACN). The desalted peptides were then eluted using 75 µL of 1.25% ammonium hydroxide (NH4OH) in 80% ACN. Eluted peptides were dried using a SpeedVac centrifuge (Eppendorf, Concentrator plus). Before reverse-phase liquid chromatography-mass spectrometry/mass spectrometry (LC-MS/MS) analysis, the peptides were reconstituted in buffer A* (0.2% TFA, 2% ACN). Optical measurement of peptide concentrations was conducted at 280 nm using a Nanodrop 2000 (Thermo Scientific), and concentrations were adjusted with buffer A* to yield a final peptide concentration of 0.5 µg/µL. A total of 1 µg of peptides was measured on a Q Exactive LC-MS/MS with a 50 cm long column (75 µm inner diameter, packed in-house with ReproSil-Pur C18-AQ 1.9 µm resin (Dr. Maisch) and a 180 min gradient using the settings stated below.

### Sample preparation of interactome samples for mass spectrometry (AP-MS)

To determine the interaction partners of the selected ISG candidates, we utilized affinity purifications followed by mass-spectrometry (AP-MS). A549 cells were transduced with lentiviral vectors carrying doxycycline-inducible V5-tagged expression cassettes for the studied ISG. Following transduction, cells were subjected to selection using 2.5 µg/mL puromycin for 3 days. The day before the experiment, 5e6 cells were seeded onto 15-cm dishes. The next day, protein expression was induced with 1 µg/mL doxycycline (Sigma-Aldrich, D5207) for 24 h. For the last 5 h, the cells were additionally pre-treated with 500 U/mL IFNα or left untreated. Subsequently, the cells were harvested as described above. The cell pellets were lysed using lysis buffer (50 mM Tris-HCl at pH 7.5, 100 mM NaCl, 1.5 mM MgCl2, 0.2% NP-40 (v/v), 5% glycerol (v/v), supplemented with cOmplete protease inhibitor cocktail (Roche, 11836170001), and 0.5% (v/v) 750 U/µL Sm DNase (MPI of Biochemistry). The lysates were then sonicated (5 min, 4 °C, 30 s on, 30 s off, low settings; Bioruptor, Diagenode). After that, protein concentrations in the cleared lysates were equalized. V5-tagged proteins were enriched by adding 50 µL of α-V5-agarose slurry (Sigma-Aldrich, A2095) and incubated with constant agitation for 3 h at 4 °C. Non-specifically bound proteins were removed through four consecutive washes with lysis buffer, followed by three detergent-removal steps with washing buffer (50 mM Tris-HCl at pH 7.5, 100 mM NaCl, 1.5 mM MgCl2, and 5% glycerol (v/v)). The enriched proteins were denatured, reduced, and alkylated, as described above. Proteins were then digested by adding 200 µL of digestion buffer (0.6 M guanidinium chloride, 1 mM TCEP (Sigma-Aldrich), 4 mM CAA (Sigma-Aldrich), 100 mM Tris-HCl at pH 8, 0.5 µg LysC (WAKO Chemicals, 4548995075888), and 0.5 µg trypsin (Sigma-Aldrich). The samples were incubated overnight at 30 °C. Peptides were desalted using C18 StageTips (Merck, 66883-U).

### LC-MS/MS measurement

Purified peptides were loaded on a 50 cm reverse-phase analytical column (75 µm diameter, 60 °C; ReproSil-Pur C18-AQ 1.9 µm resin; Dr. Maisch) and separated using an EASY-nLC 1200 system (Thermo Fisher Scientific). For peptide separation, a 120 min gradient with a flow rate of 300 nL/min was used. A binary buffer system consisted of buffer A 0.1% (v/v) formic acid in water, and buffer B 80% (v/v) acetonitrile, 0.1% (v/v) formic acid in water. More in detail, 5–30 % buffer B (95 min), 30–95 % buffer B (10 min), wash out at 95 % buffer B (5 min), decreased to 5% buffer B (5 min), and kept at 5% buffer B (5 min). Eluting peptides were directly analyzed on a Q-Exactive HF mass spectrometer equipped with a nano-electrospray source (Thermo Fisher Scientific). All measurements were performed in positive ion mode, the spray voltage was set to 2.4 kV, the funnel RF level at 60, and the heated capillary at 250 °C. Data-dependent acquisition (DDA) included repeating cycles of one MS1 full scan (300–1650 m/z, $R = 60,000$ at 200 m/z) at an ion target of 3E6 and a maximum injection time of 20 ms. For MS2 scans, the top 15 most intense peptide precursors were isolated and fragmented (R = 15,000 at 200 m/z, ion target value of 1E5, and maximum injection time of 25 ms). Dynamic exclusion, isolation window of the quadrupole, and HCD normalized collision energy were set to 20 s, 1.4 m/z, and 27%, respectively. All buffers were prepared using LC-MS grade reagents.

### MS data processing and analysis

We processed the raw MS data using MaxQuant (V 2.0.3.1) with the default settings and enabled intensity-based absolute quantification (iBAQ). The built-in Andromeda search engine was used to compare spectra against forward and reverse sequences of the reviewed human proteome, which also included isoforms obtained from Uniprot (UP000005640) and SARS-CoV-2 proteins (only in SARS-CoV-2-infected samples). For further analysis, we used the

proteinGroups.txt output file and processed the data using R. First, protein groups only identified by site, flagged as reverse sequences, or potential contaminants were filtered out. Likewise, protein groups with fewer than two unique peptides were filtered out. The remaining iBAQ log2-transformed values were then row- and column-normalized by calculating the median for each protein group (row) across all samples and then median-centering the values for each protein group by subtracting its corresponding median. Then, for each sample (column), we calculated its median, which serves as the normalization factor for each sample. For proteome data, protein groups with less than 50% missing values across all samples were then subjected to imputation utilizing the Generalized Mass Spectrum missing peaks imputation with Two-Step Lasso (GMS.Lasso) function from the GMSimpute package[112] applying the default settings. For interactome data, missing values were imputed by random numbers drawn from a shrunken (width = 0.3) and downshifted (downshift = 1.8) normal distribution derived from the distribution of the sample intensities.

To identify differentially expressed proteins, Welch's $t$-tests were applied to calculate $p$-values. Each sample was compared against its respective reference group, which consisted of all samples from the same treatment group, excluding the sample being tested: $t$-test (sample, treatment group \ {sample}) where sample $\in$ treatment group. For proteome data, every comparison had to fulfill the following criteria to be considered: less than 50% of sample values were imputed, and less than 50% of all values (sample and reference) were imputed. For interactomes, only the first criterion had to be fulfilled. Protein groups with $p \leq 0.05$ and a $|\log_2 FC| \geq 0.5$ were considered significant for proteome samples. For interactome samples, protein groups with $p \leq 0.01$ and a $\log_2 FC \geq 2$ were considered significant.

## Gene set enrichments

The functional enrichment analysis of mass spectrometry datasets was performed using g:Profiler (version e109_eg56_p17_1d3191d) with g:SCS multiple testing correction method, applying a significance threshold of 0.05 and considering all genes of the human organism in the Ensembl database as statistical domain scope[113]. We analyzed enrichments from the Gene Ontology Biological Process (GO:BP) or the Comprehensive Resource of Mammalian protein complexes (CORUM) databases[62].

## Immunofluorescence analysis

For analysis of fixed cells, A549-ACE2 BORCS8 KO cells were generated as described above. The cells were seeded on coverslips and cultured for 1 day in complete medium, then fixed for 10 min with 100% ice-cold methanol or 4 % paraformaldehyde (PFA). The coverslips were washed in PBS and blocked with 3% donkey serum, 10% fetal calf serum (FCS), and 0.1% bovine serum albumin (BSA, Sigma-Aldrich, A9418) in PBS-T. The following primary antibodies were diluted in blocking solution and incubated with the coverslips at 4 °C overnight: α-LAMP1 (Cell Signaling Technology, 9091S, 1:400), α-tubulin (Sigma Aldrich, MAB186, 1:400), α-ORF3a (Abcam, ab280953, 1:500), α-NP (Sino Biological, 40143-MM05, 1:500). The cells were washed with PBS-T and incubated with secondary antibodies: donkey anti-rabbit Alexa Fluor 488 (Invitrogen, A21206, 1:600), donkey anti-rat Cy3, (Jackson ImmunoResearch, 712-165-153, 1:600) and 4',6-diamidino-2-phenylindole (DAPI) (MCE, HY-D2868) at 1 µg/mL for 2 h at room temperature. The coverslips were washed with PBS and mounted on glass slides. The samples were imaged on a Zeiss LSM 880 with a 63x oil-immersion objective in Airyscan Fast mode or a Zeiss LSM900 with a 63x oil-immersion or a 20× objective in Airyscan 2 mode. Airyscan images were processed in Zen 2.3 (black edition, Zeiss) or Zen 3.7 (blue edition, Zeiss).For live-cell imaging, the cells were seeded on a chambered coverslip the day before the experiment. The next day, the cells were treated with the following dyes and concentrations: LysoTracker Red DND-99 (Invitrogen, L7528) at 75 nM, pHrodo Green Dextran

(Invitrogen, P35368) at 35 µg/mL, Dextran beads coupled to Alexa Fluor 647 (Invitrogen, D22914) at 10 µg/mL. Cells treated with either LysoTracker Red DND-99 or pHrodo Green were imaged using the IncuCyte S3 system as described above. Cells co-treated with pHrodo Green and Dextran AF647 were incubated for 30 min at 37 °C, washed with fresh medium, and subsequently imaged. Images were acquired every 6.5 s with an Olympus FluoView 10i (FV10i) LSM confocal microscope with a heated live-cell imaging chamber using a 60x UPLSAP waterimmersion objective. For time-resolved co-localization analysis, we isolated the green (pHrodo Green) and blue channels (Dextran AF647) for each analyzed image and applied a low-pass filter using a disc-shaped brush with a diameter of 11 pixels to minimize noise. Following noise reduction, Otsu thresholding was applied individually to each channel and time point. The colocalization between the two channels was quantified at each time by identifying pixels where both the green and blue masks were positive. The ratio of these colocalized points to the number of points positive in the blue mask (pH-insensitive dye) was calculated to obtain the colocalization ratio. The rate of endocytosis was derived from the total number of blue positive pixels.

## Double KO screening and data processing

All non-redundant combinations of the selected genes were depleted in A549-H2B-mRFP cells expressing ACE2 through sequential transduction, with each transduction followed by respective selection steps using puromycin and hygromycin resistance (Fig. 5a). Each KO was also combined with an NTC KO to represent the single KO effect. After selecting co-depleted cells, we infected each co-depletion KO cell line with a SARS-CoV-2-GFP reporter virus and acquired fluorescence images, as described above. Noise reduction was first achieved by omitting time points preceding the initial local minimum. Subsequently, a baseline correction was performed on each measurement series, normalizing the signal values to the first time point. For more reliable trend analysis, data smoothing was carried out using a Locally Estimated Scatterplot Smoothing (LOESS) algorithm. The ratio of green integrated intensity (SARS-CoV-2-GFP) over H2B-mRFP (cell number) of each co-depletion—denoted here by $A$ and $B$—at $t = 48$ h was then extracted and normalized by dividing the values by the median effect of the corresponding single KO within each replicate. We estimated the expected double KO (dKO) effect via Eq. (5), where $KO_A$/NTC and $KO_B$/NTC represent the observed effects of the individual gene KOs.

$$dKO_{expected} = \frac{KO_A}{NTC} \cdot \frac{KO_B}{NTC} \qquad (5)$$

The individual KO effects are multiplied in this equation to project the effect of $KO_A$ and $KO_B$. We subsequently compute the interaction score (IC) by subtracting the expected value from the observed effect of the dKO using Eq. (6).

$$IC = dKO_{observed} - dKO_{expected} \qquad (6)$$

This interaction score quantifies the deviation of the combined effect of the two KOs from their anticipated multiplicative effect, as inferred from individual KO effects. The interaction score thus measures the relative impact of the dKOs compared to either single KO effect. A positive interaction score suggests that the combined effects of the two KOs are greater than the expected multiplicative effect of the individual KOs. This could indicate that the two genes operate in different pathways or stages of the viral life cycle, and their combined KO has a synergistic effect. Conversely, a negative interaction score suggests that the combined effect of the dKO is less than the expected effect. This typically indicates redundancy, where the two genes likely

operate within the same pathway or contribute to the same stage of the viral life cycle.

## Bioinformatic analysis and code availability

If not otherwise specified, the data were analyzed using R (version 4.2.2, 2022-10-31 ucrt) and RStudio (version 2023.06.1, Build 524). The following R packages and versions were used: broom (1.0.5), circlize (0.4.15), ComplexHeatmap (2.14.0), EBImage (4.40.1), ggpubr (0.6.0), ggrepel (0.9.3), gprofiler2 (0.2.2), ggsignif (0.6.4), glmnet (4.1-8), GMSimpute (0.0.1.0), heatmaply (1.4.2), kohonen (3.0.12), pheatmap (1.0.12), rstatix (0.7.2), splines (4.2.2), tidyverse (2.0.0). The scripts for curve fitting of IncuCyte image data are available at https://github.com/innatelab/isg-atlas.

## Reporting summary

Further information on research design is available in the Nature Portfolio Reporting Summary linked to this article.

## Data availability

An interactive version of the virus-host interaction dataset is available at https://isg-atlas.innatelab.org. All data are included in the Supplementary Information or available from the authors, as are unique reagents used in this Article. The raw numbers for charts and graphs are available in the Source Data file whenever possible. The mass spectrometry proteomics data have been deposited to the ProteomeXchange Consortium via the PRIDE partner repository[114]. This includes the following datasets: proteome of RTCB KO cells (PXD045916), affinity purification of cells overexpressing RTCB (PXD045929); proteomes of ISG KOs (PXD045783) and affinity purification of overexpressed ISGs (PXD045812). Source data are provided with this paper.

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

## Acknowledgements

We extend our gratitude to Aleksandra Babnis, Claire Burny, Søren Riis Paludan, and Virginie Girault for their critical review and constructive feedback on the manuscript. We thank Beate Sodeik for recombinant herpesviruses and Joachim Bugert for VACV-GFP. We also thank Robert Baier for his assistance with the cloning of overexpression constructs and Christian Urban for his support in analyzing the proteomics data. Moreover, we thank Diego for his support throughout this research. The work in A. Pic.'s laboratories was funded by an ERC Consolidator grant (ERC-CoG "ProDAP", 817798), the European Union (EU4Health grant "NoVir"; 101203301), the German research foundation (TRR179/TP10, TRR237/A07 and TRR353/B04), the Helmholtz Association's Initiative and Networking Fund ("COVIPA"; KA1-Co-02), the Bavarian State Ministry of Science and Arts (Bavarian Research Network "FOR-COVID" and "Prevention of Pandemic-infection-associated Pathology Munich - P3M"; BayVFP 2024-2027), and the Danish National Research Foundation (CiViA; DNRF164). J.R.B. was funded by a Marie Skłodowska-Curie Actions Postdoctoral Fellowship ("Paris"; 101107996).

## Author contributions

K.K. and A.Pic. conceptualized the study. A.Pic. acquired funding and supervised the study. K.K., J.R.-B., S.H., S.M., S.B., Q.E., M.V., S.Mu., A.C.,

A.P., and V.G. investigated the study. K.K. and J.R.-B. validated the data. K.K., S.H., M.V., J.R.-B. and V.B. performed formal analysis. K.K., and S.H. developed the methodology. K.K., S.H., and J.R.-B. performed visualization. K.K. and A.Pic. wrote the original draft. K.K., J.R.-B., S.H., S.B., Q.E., S.Mu., V.B., V.G., and A.Pic. reviewed and edited the manuscript.

## Funding

## Competing interests
The authors declare no competing interests.
