## [Transparent Peer Review file · Nature Communications]

The ISG Atlas: A Loss-Of-Function Analysis Characterizes Antiviral Properties of Interferon Stimulated Genes

Corresponding Author: Dr Andreas Pichlmair

Version 0:

Reviewer comments:

Reviewer #1

(Remarks to the Author)

In their manuscript, "The ISG Atlas: A Loss-Of-Function Analysis Characterizes Antiviral Properties of Interferon Stimulated Genes," Krey et al explore the functional activity of hundreds of interferon-stimulated genes (ISGs) on eight different viruses. The manuscript is founded on a time-resolved, loss-of-function screen, in which ISGs are individually disrupted by CRISPR-Cas9 in A549 cells, which are then stimulated with Type I Interferon (IFN). Cultures are then infected with different reporter viruses, and reporter intensities relative to control reporter signal (i.e. A549 cells) are measured over time and modeled to quantify the effects of ISG disruption on viral replication. Results of this large-scale screen include numerous antiviral and proviral ISGs for the panel of viruses tested. A subset of screen hits are investigated further in proteomics experiments (investigating potential interactions), and dual gene knockout experiments (investigating potential synergies). Additional functional experiments expanded on proviral roles for RTCB and antiviral roles for BORCS8 in SARS-CoV-2 infection.

Although prior work from many research groups have utilized different screening technologies to explore the antiviral activities of IFN, the activities of ISGs and how they may function together against different viruses remains incompletely understood and a significant research question. The present study makes a meaningful contribution to these efforts and provides a resource likely to be of considerable utility. In particular, performing the loss-of-function screens in the context of an active IFN response offers a complementary approach to prior overexpression and related ISG screening approaches. Furthermore, incorporating longitudinal timepoint measurements to assess viral restriction represents a strength of the experimental design, and is likely to uncover phenotypes that may not have been apparent if measuring a single "snapshot" of the infection process. Additional studies, including paired gene knockout experiments and proteomics exploration of potential interactions further enhance the work. However, there remain some questions and concerns regarding data interpretations, normalization strategies, and aspects of functional experiments. These and additional comments are listed below.

MAJOR POINTS

As noted above, the general design and implementation of the screen, incorporating loss-of-function tests in the context of an active IFN response over time is a strength of the manuscript. That said, there are some concerns regarding the interpretation of screen results that at minimum require clarification, and/or potential additional consideration or analysis.

-The ISG screen design utilizes CRISPR-Cas9 to disrupt expression of target ISGs. However, no data are provided regarding the efficiency of gene knockout. Presumably, each target-ISG culture is heterogeneous with regard to gene knockout, and this heterogeneity may vary across target ISGs. While this may be an inherent technical limitation of this approach, the authors should at minimum address how such heterogeneity might affect results, particularly with regard to comparisons across different ISGs for antiviral activity. Furthermore, though unlikely practical for the entire library, data demonstrating "typical" knockout efficiency across a subset of ISGs could be used to better support the approach.

-The authors include a "host cell" reporter (GFP or mRFP tagged H2B) against which to normalize virus-reporter signal. Details on how these normalizations were performed and interpreted appear in the Methods section (lines 653-657). However, it is somewhat unclear whether normalization was performed per cell (e.g. using a segmentation analysis) or per total fluorescent signal per well (which seems to be indicated by "integrated signal" of positive objects). Assuming the latter, it is unclear how to interpret virus restriction parameters (e.g. K) with the possibility of cytopathic effects of the infection. For

example, it seems possible, particularly in the context of a potentially heterogeneous (i.e. gene knockout) culture that a knockout resulting in rapid cell death upon infection could result in a rather low viral fluorescence:host cell fluorescence ratio, when in fact, the target ISG provides potent restriction. Other scenarios that could lead to misinterpretation are also possible. For example, APOBEC3A is highlighted as a proviral hit. APOBEC3A has known pro-apoptotic effects; could this 'proviral' effect be a consequence of improved cell survival?

The follow up studies on BORCS8 restriction of SARS-CoV-2 further endorse the screening methodology and add to the manuscript. However, data presented in the manuscript raise some additional questions.

- The experiments presented support a role for BORCS8 acting at endosomal entry of SARS-CoV-2. A restrictive effect is also observed in Calu3 cells, which unlike A549 cells express TMPRSS2 (thereby potentially bypassing an endosomal entry route). The proteomics data suggest notable interactions with ribosome components and the protein translation machinery; this is underexplored as a potential additional mechanism for viral restriction.

- BORCS8 knockout validation studies would be enhanced by a "rescue" phenotype with ectopic BORCS8 expression

The dual ISG synergy analysis emphasizes the cooperative ISG functions the interferon-induced antiviral state and is an additional strength of the manuscript. Points for clarification include:

- Related to the above point regarding CRISPR-Cas9 knockout efficiency, it seems possible and/or likely that for each ISG pair, a culture may contain cells with both ISGs disrupted, one or the other ISG disrupted, or neither disrupted. Is this taken into account in comparing across pairs and/or assessing phenotypes more generally?

- The authors might consider an alternative designation for negative scored pairs. "Antagonistic" does not seem appropriate for pairs like STAT2 and IRF9, which act cooperatively in the ISGF3 complex.

ADDITIONAL POINTS

- In Figure 2A, for most viruses, there are approximately equal amounts of "antiviral" and "proviral" ISGs. While ISGs are known to have both effects, it is somewhat surprising/unexpected that "proviral" ISGs would be so relatively abundant. The authors might discuss this further.

- Figure 2B: A beeswarm plot might improve visualization of the individual ISGs and accompanying labels

- The BORCS8 experiments presented in Figure 5 A and B show a somewhat more modest effect on viral replication than might be expected from other data presented. The authors might consider single cycle (high multiplicity of infection) and multi-cycle (low multiplicity of infection) viral growth curves to better assess the magnitude of the BORCS8 effects.

- The proteomics interactome data provides rich data that may suggest mechanisms of action for ISG functions. While the enrichment terms provide some possible clues, was any additional analysis of specific protein interactions performed? Is there any evidence of ISG interaction with viral proteins?

- In the Results section, lines 140-174: while highlighting screening results here seems appropriate, in the absence of specific validation studies, some of the more speculative points about ISG function might be more appropriate in the Discussion section.

- Throughout the manuscript, certain terminology regarding what is measured/quantified in the microscopy data should be modified. For example, the authors refer to "viral spread" but unless frequencies of infected cells were quantified by segmentation analysis or similar, this may not appropriately describe the data available.

- Relatedly, terminology such as "antiviral activities" implies direct effector functions. At Lines 127-129, counting STAT1 and IRF9 among ISGs with broad "antiviral activities" is therefore somewhat inexact; these are transcription factors that activate expression of all other ISGs (with antiviral activities). The same likely applies for DDX58 (Line 132), which likely acts as a viral sensor rather than effector.

- Line 51: "curtail" may not be the correct word here

- Line 71: use of "necessary" instead of "crucial" may align better with common terminology

Reviewer #2

(Remarks to the Author)

The interferon (IFN) response is a highly potent anti-viral defense mechanism. Upon exposure of cells to IFN, hundreds of IFN stimulated genes (ISGs) are regulated, many of which known anti-viral factors. However, the virus species-specific impact of individual ISGs and their contribution to the overall anti-viral effect of IFN is still somewhat enigmatic.

In this manuscript Krey and colleagues present a comprehensive overview of the contribution of ~300 ISGs to the anti-viral impact against 8 different viruses. They identify common anti/pro viral as well as virus-species specific ISGs and cluster the response patterns using machine learning. Finally, they functionally characterize two selected hits, RTCB and BORCS8.

Overall, this is a very impressive, highly interesting and relevant collection of data, which will certainly also inspire future compelling research. Generally speaking the novel conclusions of the manuscript are fully supported by the presented data. Especially ISG vs virus map and corresponding analyses are very well planned and carried out, and provide significant insight in pan pro/anti-viral and specific pro/anti-viral impact of ISGs. However, the mechanistic analyses fall a bit short of explaining why the chosen candidate ISGs promote/restrict viruses and thus should be strengthened.

Major:

One important question is always, which ISGs are relevant for the IFN response against a specific virus i.e. what is a 'minimum' set of ISGs. Starting from the overview in Fig1 and Fig2 as well as the co-depletion analyses, can the authors define (I) which ISGs are likely to e.g. contribute 95% of the anti-viral effect of IFN α . (II) Calculate the contribution of synergistic active ISGs to the whole IFN-mediated restriction of SARS-CoV-2. Furthermore, could all machine learning neurons be clustered to a overarching principle or gene ontology term, and in the results section described which complexes are known, which are not (extending the current description)?

While the functional analyses of RTCB and BORCS8 provide a lot of detail about their interaction partners, the molecular mechanism how they impact the respective viruses is still unclear. I suggest that the authors add experimental evidence of the impact of one of these factors on viral replication to strengthen the manuscript. For example, does depletion of BORCS8 increase fusion/entry of incoming virions (as the pseudotype assays would hint at)? Does ORF3a expression alone alter BORCS8 localisation patterns, hinting at a hijacking of BORCS8 by the viral accessory protein?

Minor:

Any virus-assay based solely on cell line data usually faces the criticism, whether the results would be the same in a more relevant setting i.e. primary cells. While I do understand that the main focus of this manuscript is to provide an 'atlas', it would inspire more confidence in the screen if a few selected (extreme) hit from the groups of pan pro/anti-viral and specific pro/anti-viral were confirmed in primary human lung cells.

One bottleneck of CRISPR screening approaches is always the efficiency of the gRNA. Can the authors briefly comment (in the results section) on how the (pools of) gRNAs were selected and whether/how the KO efficiency in A549 was quantified to allow the conclusion that each gRNA targets each ISGs efficient enough.

The authors may want to revisit their statements about TRIM25 and ZC3HAV1, also known as ZAP. TRIM25 is a rather well-known co-factor of ZAP, thus the results of the authors do not uncover a novel connection but rather confirm a well-known one within their data set. Compare: Li et al, Plos Path, 2017; Zheng et al, JVI, 2017; Nchioua et al, mBio, 2020.
line 400: ORF3a ?

Reviewer #3

(Remarks to the Author)

In this study, the authors systematically investigated 285 ISGs for their virus-modulating activity against eight viruses by utilizing a time resolved, arrayed loss-of-function screen. This comprehensive analysis identified several ISGs with yet unreported virus specificity and ISGs with pan-antiviral activity, such as RNA 2',3'-cyclic phosphate, 5'-OH ligase (RTCB). Furthermore, the authors identified BORCS8 which had a particularly prominent role in modulating SARS-CoV-2 infection by mediating the acidification of early endosomes during the viral entry, a process known to facilitate virus particle degradation. Overall, this study is potentially interesting. However, the novelty and depth of research need to be improved. Most of the findings about antiviral activity of various ISGs have been previously reported. BORCS8 as a regulator of SARS-COV-2 infection is only supported by in vitro studies. Other concerns are listed below.

1. The authors used only one cell line (A549) to evaluate the antiviral activity of the tested ISGs on various viruses, including Herpes simplex virus 1 (HSV-1) strain 17+, Measles virus (MeV) strain EdTag, Rift Valley Fever Virus (RVFV) ZH548 strain, Severe acute respiratory syndrome coronavirus 2 (SARS-CoV-2) MUC-IMB-1 strain, Semliki Forest virus (SFV) SFV6 strain, Vaccinia virus (VACV) WR strain (V300), Vesicular stomatitis virus (VSV) Indiana strain, and Yellow Fever virus (YFV) 17D strain. Different types of cells should be tested to confirm their findings.
2. Most of the findings in this study is data-driving without experimental examinations. For example, in Figure 4, the synergistic effects of several ISGs, such as STAT2, LY6E, BORCS8 et al., should be confirmed by experiments in double- or triple-knockout cells.
3. The authors could introduce other databases such as Biogrid and String to analysis intercoms of candidate ISGs, which might provide additional insights for understanding how these ISGs function individually or synergistically.
4. In Figure 5, the authors found that BORCS8 mediated the acidification of early endosomes to regulate SARS-COV2 infection, which should be confirmed by more experimental evidences. Furthermore, the molecular mechanisms need to be explored.

Version 1:

Reviewer comments:

Reviewer #1

(Remarks to the Author)

This revised manuscript, "The ISG Atlas: A Loss-Of-Function Analysis Characterizes Antiviral Properties of Interferon Stimulated Genes," from Krey et al represents a significant improvement from the initial submission. In general, the authors were responsive to reviewer comments, particularly with regard to mechanistic and follow up experiments involving BORCS8. Overall, as noted in the initial review and now further improved upon in the present submission, this manuscript remains highly significant and an impressive body of work. It is likely to be of considerable interest to the virology and innate immunity fields. As previously acknowledged, the strengths of the study remain the innovative experimental design (testing ISG loss of function in the context of an IFN response), the breadth of different viruses and ISGs evaluated, and the longitudinal timepoints to assess viral restriction. However, with regards to the ISG screen, the central focus of the manuscript, concerns remain about data interpretation and presentation, particularly with regard some of the quantitative comparisons made throughout the manuscript.

To state once again, the ISG/virus screen is very impressive and represents an important advance. However, it is clear both from the updated text and the responses to initial review that the screen (necessarily) includes many sources of heterogeneity (detailed below) across ISG and virus conditions. This raises significant concerns about how the data are interpreted and described throughout the manuscript; the authors frequently refer to certain ISGs as "more antiviral" than others, or ascribe "strong" pro/antiviral activities to certain ISGs, which may not be appropriate given the (understandable) technical limitations of the approach.

There exist several sources of technical heterogeneity in the experimental design of the ISG screen:

-In a welcome response to the initial review, the authors provided qPCR data for CRISPR knockout efficiency for a subset of target ISGs. These data, presented in Supplemental Figure 1, demonstrate CRISPR-mediated knockout for all ISGs tested. However, the results also further emphasize that the knockout efficiency is considerably variable across ISGs.

-In an additional response to initial review, the authors helpfully clarified the normalization approach used in image analysis of the screen; specifically, "The normalization in our analysis was performed per image by calculating the ratio of viral fluorescence to host cell fluorescence, which we found to be most robust and led to the most reproducible results." This seems an appropriate strategy, particularly for an arrayed screen of this size (the authors note that thousands of images were analyzed). However, it introduces an additional source of variance that is likely to vary ISG-to-ISG. Whole-image fluorescence intensity does not necessarily represent "viral propagation." One can imagine a single infected cell that increases fluorescence intensity over time. This might provide a similar signal to one cell maintaining fluorescence intensity but spreading infection to many adjacent cells. While not necessarily critical in designating an ISG as "anti" or "pro" viral, these differences could have implications for the mode of viral inhibition. Responsive to the initial review, the authors appropriately removed the descriptor "spread" from the manuscript (which cannot be evaluated without measure infection at single cell resolution). However, without a measure of infectious viral particles, "propagation" is similarly inappropriate (e.g. a non-propagating infected cell could increase fluorescence intensity over time).

- Infections were performed at low and high MOIs, depending on virus. This is an appropriate design choice, given differences in virus replication rate, cytopathogenicity, and fluorescence intensities. However, it again makes it challenging to compare ISGs quantitatively, further compounded by the fluorescence normalization strategy highlighted above.

In summary, the above sources of variation make quantitative comparisons of ISGs particularly challenging. To be abundantly clear, these technical limitations do not invalidate a very impressive and important collection of screens; as noted previously this is a very impressive and informative body of work and should be recognized as such! The ISG effects, particularly the antiviral effects, detected appear robust and indicative of important viral restriction functions. However, the authors are encouraged to be extremely cautious with their interpretations and related language regarding quantitative comparisons of ISG activities. It is appreciated that, appropriately, many of the ISGs are described as "broadly" acting or similar, which is appropriate and non-quantitative. However, this is not universal throughout the manuscript. And while it is further appreciated that the authors added some language describing potential variation of knockout efficiency to the Discussion section (lines 497-502), there remains insufficient discussion of how this and the above-described issues should affect interpretation of the data.

In addition, related to the above: if K and tau values can be impacted by the highlighted sources of variation (particularly CRISPR knockout efficiency), how might using these values as input to the SOM impact results and interpretation?

An additional consideration in the screen design is the potential impact of cell growth rates. The authors are quite rigorous with their assessment of impacts of ISG knockout on cell growth rates. However, it seems that the effects of knockout on cell growth were tested in the absence of IFN. This could be problematic because 1) Experiments were conducted in the presence of IFN. Experiments were conducted across 48-72 hours, which is more than enough time for >1-2 replicative cycles of A549 cells 2) The knocked out genes are ISGs, so they may not be expressed in the absence of IFN / potential effects on proliferation not apparent in absence of IFN, 3) IFN is known to affect cell proliferation (as acknowledged in the manuscript text). The authors should address this point and how it was managed in screen analyses.

Additional minor points:

Lines 48-49: "To maximize the efficacy of the cellular defense system, the host has expanded its repertoire of ISGs," This sentence is unclear. Do the authors mean evolutionarily expanded? If so, clarifications (and citations would be appropriate).

In the Supplementary Table 2 “categorized data” tab, category labels are rather confusing. With STAT1 and YFV as a “simple” example, the K_category is “2_major_proviral” (K = 8.593) which suggests that KNOCKOUT of STAT1 PROMOTES YFV (“proviral”) but tau_category is “2_major_slower” (tau = 6.227) which suggests that KNOCKOUT of STAT1 SLOWS YFV, but this is not correct...knockout of YFV promotes the speed of YFV as indicated on the Fig 1 heatmap. As I interpret the category labels, the “directions” for K and tau categories are “inverted” to each other.

BORCS8 expression was induced with Doxycycline. Were +/- doxycycline controls included in these experiments (as doxycycline can have effects on viral replication and/or cell proliferation)?
Figure 1 lists panels “a” and “c” but no panel “b”.

In the Results text, what was considered when designating “pro/anti” viral? K or tau? It is not entirely clear as presently written.

Reviewer #2

(Remarks to the Author)

The authors have addressed all my concerns. Thanks a lot and great work!

as the mediator for reviewer #3

Reviewer 3 had four major issues:

1. Only one cell line was used by the authors to characterise the impact of KO/viral infection - The authors reasoned why the cell line in question was used, but now also additionally provided data in a second cell type (even primary cells). From my point of view that comment has been addressed.
2. Data-driving without experimental examinations - I agree, the foundation of (empiric) scientific progress is the combination of data collection and data interpretation. Thus, collecting data without interpretation is practically meaningless, and provides no conceptual advance. However, I do not agree that the presented manuscript is such an empty collection of data. The screens were verified, combinatorial effects looked at and mechanistic experiments performed. In addition, the data is interpreted, set in context and provides -according to my opinion- a significant advance.
3. The authors should introduce other database – This was addressed by the authors. I would agree that a full integration of IFN stimulated/non stimulated datasets is highly interesting but beyond the scope of the current study.
4. More mechanistic analyses are required for BORCS8 – The authors have provided more mechanistic data. This was in line with the answers given to the other reviewers.

According to my opinion, the authors have addressed all the issues raised by reviewer 3 sufficiently, and the manuscript has significantly improved.

REVIEWER COMMENTS

Reviewer #1 (Interferon responses/ISG)(Remarks to the Author):

In their manuscript, “The ISG Atlas: A Loss-Of-Function Analysis Characterizes Antiviral Properties of Interferon Stimulated Genes,” Krey et al explore the functional activity of hundreds of interferon-stimulated genes (ISGs) on eight different viruses. The manuscript is founded on a time-resolved, loss-of-function screen, in which ISGs are individually disrupted by CRISPR-Cas9 in A549 cells, which are then stimulated with Type I Interferon (IFN). Cultures are then infected with different reporter viruses, and reporter intensities relative to control reporter signal (i.e. A549 cells) are measured over time and modeled to quantify the effects of ISG disruption on viral replication. Results of this large-scale screen include numerous antiviral and proviral ISGs for the panel of viruses tested. A subset of screen hits are investigated further in proteomics experiments (investigating potential interactions), and dual gene knockout experiments (investigating potential synergies). Additional functional experiments expanded on proviral roles for RTCB and antiviral roles for BORCS8 in SARS-CoV-2 infection.

Although prior work from many research groups have utilized different screening technologies to explore the antiviral activities of IFN, the activities of ISGs and how they may function together against different viruses remains incompletely understood and a significant research question. The present study makes a meaningful contribution to these efforts and provides a resource likely to be of considerable utility. In particular, performing the loss-of-function screens in the context of an active IFN response offers a complementary approach to prior overexpression and related ISG screening approaches. Furthermore, incorporating longitudinal timepoint measurements to assess viral restriction represents a strength of the experimental design, and is likely to uncover phenotypes that may not have been apparent if measuring a single “snapshot” of the infection process. Additional studies, including paired gene knockout experiments and proteomics exploration of potential interactions further enhance the work. However, there remain some questions and concerns regarding data interpretations, normalization strategies, and aspects of functional experiments. These and additional comments are listed below.

We thank Reviewer #1 for highlighting the strengths of the study, including the innovative application of loss-of-function screens in the context of an active IFN response and the integration of longitudinal time-point measurements. The detailed suggestions regarding data interpretation, normalization methodologies, and the functional experiments have been carefully considered, and the feedback has provided clear guidance for refining the analyses and reinforcing the conclusions presented.

MAJOR POINTS

As noted above, the general design and implementation of the screen, incorporating loss-of-function tests in the context of an active IFN response over time is a strength of the manuscript. That said, there are some concerns regarding the interpretation of screen results that at minimum require clarification, and/or potential additional consideration or analysis.

-The ISG screen design utilizes CRISPR-Cas9 to disrupt expression of target ISGs. However, no data are provided regarding the efficiency of gene knockout. Presumably, each target-ISG culture is heterogeneous with regard to gene knockout, and this heterogeneity may vary across target

ISGs. While this may be an inherent technical limitation of this approach, the authors should at minimum address how such heterogeneity might affect results, particularly with regard to comparisons across different ISGs for antiviral activity. Furthermore, though unlikely practical for the entire library, data demonstrating “typical” knockout efficiency across a subset of ISGs could be used to better support the approach.

We agree with the reviewer that heterogeneity is an inherent technical limitation of large-scale screens. To get a better understanding of this heterogeneity and the knockout (KO) efficiency, we performed additional RT-qPCR experiments to assess KO efficiencies for a subset of ISGs, now included as Supplementary Figure 1d. These data show substantial reductions in ISG transcript levels in the respective KO cells, demonstrating their efficient depletion in KO cells. IRF9 KO cells, which exhibited superior growth of many viruses, showed a 65% reduction in IRF9 mRNA. Similarly, OAS1 depletion promoted the growth of multiple viruses, and OAS1 mRNA was reduced by 80% in the KO cells as compared to controls. Additionally, proteomics data from Supplementary Figure 3c shows significant reductions in protein abundance for all targeted ISGs that could be detected by mass spectrometry as compared to non-targeting (NTC) control cells. Together, these results confirm effective disruption at both transcript and protein levels. We also evaluated cell growth rates and removed 15 cell lines that showed reduced growth compared to controls to avoid bias due to cell density. While it is challenging to assess knockdown heterogeneity across the entire dataset, the differential growth behavior of viruses observed in KO cells suggests a functional depletion of the respective ISGs. We have also added paragraphs in the Results and Discussion section, acknowledging the potential impact of heterogeneity in KO efficiency on phenotypic comparisons and emphasizing that trends across replicates and independent experiments mitigate these concerns (lines 105-105 and 496-498).

-The authors include a “host cell” reporter (GFP or mRFP tagged H2B) against which to normalize virus-reporter signal. Details on how these normalizations were performed and interpreted appear in the Methods section (lines 653-657). However, it is somewhat unclear whether normalization was performed per cell (e.g. using a segmentation analysis) or per total fluorescent signal per well (which seems to be indicated by “integrated signal” of positive objects). Assuming the latter, it is unclear how to interpret virus restriction parameters (e.g. K) with the possibility of cytopathic effects of the infection. For example, it seems possible, particularly in the context of a potentially heterogeneous (i.e. gene knockout) culture that a knockout resulting in rapid cell death upon infection could result in a rather low viral fluorescence:host cell fluorescence ratio, when in fact, the target ISG provides potent restriction. Other scenarios that could lead to misinterpretation are also possible. For example, APOBEC3A is highlighted as a proviral hit. APOBEC3A has known pro-apoptotic effects; could this ‘proviral’ effect be a consequence of improved cell survival?

We agree that normalization strategies can significantly affect the interpretation of virus restriction parameters, especially in cases when multiple factors with distinct dynamics are compared. During the setup of this screen, we found that normalization for cell density is a crucial factor in ensuring comparability between different conditions and experimental batches, as well as in guaranteeing similar infection rates. The normalization in our analysis was performed per image by calculating the ratio of viral fluorescence to host cell fluorescence, which we found to be most robust and led to the most reproducible results. This approach enables high-throughput analysis across thousands of images. Similar principles are used for other experimental setups (e.g. western blotting) where loading controls are used to ensure similar protein content between experimental groups. Reviewer 1 asks the interesting question of whether the values we estimate

in the modelling may be affected by the normalization to cell density in a KO and virus-dependent context. To minimize the risk that differential cell density leads to artificial misinterpretation of virus growth, we ensured that all tested ISG KO cells had at least 80% growth rate of control cells as observed in a time frame of 72 h. As reviewer 1 points out, virus-induced cell death could also affect the analysis. To evaluate the influence of virus infection on normalization, we randomly selected 100 KO cells and evaluated the cell fluorescence values at the time point of the maximal virus signal. This analysis suggests that the cell fluorescence values were

relatively stable across different KO cells and for the time considered in this experiment (Figure R1), indicating that the computed K values were not massively affected by variability in the cell fluorescent signal. If cell fluorescence signals would indeed be reduced due to increased virus-induced cell death, the analysis would still evaluate the ISG as “antiviral” and flag it as a potentially relevant ISG, which is the main purpose of this screen.

The follow up studies on BORCS8 restriction of SARS-CoV-2 further endorse the screening methodology and add to the manuscript. However, data presented in the manuscript raise some additional questions.

- The experiments presented support a role for BORCS8 acting at endosomal entry of SARS-CoV-2. A restrictive effect is also observed in Calu3 cells, which unlike A549 cells express TMPRSS2 (thereby potentially bypassing an endosomal entry route). The proteomics data suggest notable interactions with ribosome components and the protein translation machinery; this is underexplored as a potential additional mechanism for viral restriction.

While our proteomics data indicate interactions between BORCS8 and components of the ribosome and translation machinery, our functional studies prioritized BORCS8’s established role in endosomal trafficking and lysosomal function, which we demonstrated as critical for its antiviral activity. Reviewer 1 correctly mentions that Calu-3 cells express TMPRSS2. However, in our hands cells that express TMPRSS2 (Calu3, primary human bronchoepithelial cells) endogenously are still partially sensitive to inhibitors of the endosomal pathway (e.g. BB96, E64D)

inhibitors (Jocher et al., EMBO reports 2022), indicating an additional role in endosomal uptake, which is in line with an increase in virus infection in the absence of BORCS8.

To further show that BORCS8 depletion indeed affects endosomal uptake, we transcomplemented A549 cells with TMPRSS2 and tested for the effect of BORCS8 depletion on virus growth. While TMPRSS2 increased virus growth in A549 cells, depletion of BORCS8 did not additionally affect virus replication (Figure 5e). This indicates that forcing plasma membrane fusion by overexpression of high levels of TMPRSS2 overcomes the inhibitory effect of BORCS8, further supporting a role of BORCS8 in viral uptake.

- BORCS8 knockout validation studies would be enhanced by a “rescue” phenotype with ectopic BORCS8 expression

We agree with reviewer 1 that rescue experiments for the BORCS8 KO are important to show specificity. Thus, we re-constituted BORCS8 KO with a codon-optimized BORCS8 construct and found that this reduced viral replication to levels comparable to those observed in wild-type control cells (Figure 5c). Overall, this rescue experiment validates the antiviral effect of BORCS8 and confirms the specificity of the BORCS8 depletion in KO cells.

The dual ISG synergy analysis emphasizes the cooperative ISG functions the interferon-induced antiviral state and is an additional strength of the manuscript. Points for clarification include:

- Related to the above point regarding CRISPR-Cas9 knockout efficiency, it seems possible and/or likely that for each ISG pair, a culture may contain cells with both ISGs disrupted, one or the other ISG disrupted, or neither disrupted. Is this taken into account in comparing across pairs and/or assessing phenotypes more generally?

The double KOs were generated through sequential dual transduction with individual positive selection, ensuring that all cells were transduced with both lentiviruses. While some cells may retain wild-type protein expression due to indels that do not cause functional disruption, the majority of cells are expected to carry disruptions in both genes, as supported by our data on typical KO efficiency. Functional targeting is further supported by the strong effects observed in the double KO screening - often exceeding the effects mathematically expected from the combination of the two single KOs. Additional support for KO efficacy comes from the observation of similar effects for similar genes – i.e. similarities between the well-characterized ISG LY6E and the here described candidate BORCS8, revealing a similar mechanism at play, which further supports the robustness of our observations. Altogether, this double-KO screen tested for 400 knockout combinations – systematically showing successful depletion of all target gene combinations was unfortunately not feasible – we ask reviewer 1 for his understanding.

- The authors might consider an alternative designation for negative scored pairs. “Antagonistic” does not seem appropriate for pairs like STAT2 and IRF9, which act cooperatively in the ISGF3 complex.

We agree with reviewer 1 and removed the term "antagonistic". We now refer to non-additive effects, or effects that are less than the expected product of ISG co-depletion.

ADDITIONAL POINTS

-In Figure 2A, for most viruses, there are approximately equal amounts of “antiviral” and “proviral” ISGs. While ISGs are known to have both effects, it is somewhat surprising/unexpected that “proviral” ISGs would be so relatively abundant. The authors might discuss this further.

We fully agree with reviewer 1 that the number of proviral ISGs identified by our screening method is surprising. We clarified in the results that approximately one-quarter of the tested ISG KOs per virus exhibited minor pro- or antiviral activity. We feel that the activity of a small set of antiviral ISGs may dominate over the partially mild pro-viral activities of other ISGs. We expanded the discussion on the unexpected abundance of proviral ISGs, emphasizing their potential roles in cellular processes that viruses may exploit. This underscores the complexity of ISG functions and their dual impact on host-virus interactions.

-Figure 2B: A beeswarm plot might improve visualization of the individual ISGs and accompanying labels

As suggested by reviewer 1, we added a beeswarm blot in Figure 2b.

-The BORCS8 experiments presented in Figure 5 A and B show a somewhat more modest effect on viral replication than might be expected from other data presented. The authors might consider single-cycle (high multiplicity of infection) and multi-cycle (low multiplicity of infection) viral growth curves to better assess the magnitude of the BORCS8 effects.

We have compared the effect of BORCS8 KO to the effect of LY6E KO, which is a well-accepted inhibitor of SARS-CoV-2. In Incucyte experiments, we can find similar effects indicating a prominent effect of BORCS8. However, as we show in Figure 5, BORCS8 appears to regulate viral uptake as well as exocytosis highlighting a dual effect in the viral life cycle.

-The proteomics interactome data provides rich data that may suggest mechanisms of action for ISG functions. While the enrichment terms provide some possible clues, was any additional analysis of specific protein interactions performed? Is there any evidence of ISG interaction with viral proteins?

In this study, we focused on the functional classification of ISGs and their downstream effects and intersected GO:BP terms across four proteomic datasets to derive functional insights. While this analysis gives a broad overview of the activity of the tested ISGs, it was not possible to obtain in-depth insights on the activity of the tested ISGs. However, a subset of the proteins identified in the interactome analysis and the enriched GO:BP terms are in part also identified in virus-host interactome screens.

Inspired by the request of Reviewer 1 we intersected the interactomes of ISGs, including BORSC8, with interactors of SARS-CoV-2 proteins for which a comparable AP-MS strategy was used (Stukalov et al., 2021). We found that there was only a limited number of overlapping proteins ($n = 32$) (Figure R2a). Although this overlap is interesting, it is difficult to extract mechanistic insights from this intersection, particularly since most viral and ISG proteins share only 2-3 interactors and it is unclear whether these interactions are specific for SARS-CoV-2 – ISG interactions or whether they would also occur in other viral datasets. However, a systematic overlap of ISG interactors with interactors of viral proteins suffers from unavailability of appropriate virus interactome datasets that were analysed in a manner that is comparable to the ISG-interaction analysis. Interestingly, proteins that are changing in abundance after ISG expression (ISG effectomes, Supplementary Figure 3d) were often found as interactors of viral proteins (Figure R2b). This may indicate that ISG regulated proteins are relevant for viral propagation. However, this hypothesis would require rigorous testing and intersection with additional virus-host interaction datasets that are currently not available, as mentioned above. Thus, although such intersections are highly interesting, we ask reviewer 1 for his/her understanding that we would opt not to add these data to the manuscript since we would like to avoid making claims or suggestions that cannot be well-controlled in this manuscript.

Figure R2: Overlap of proteins of ISG functions with SARS-CoV-2 interactors. **(a)** Intersection of proteins interacting with with SARS-CoV-2 proteins with proteins interacting with 20 ISGs that were identified to affect SARS-CoV-2 and were analysed in this study. **(b)** Intersection of SARS-CoV-2 interacting proteins with proteins regulated by ISG expression.

-In the Results section, lines 140-174: while highlighting screening results here seems appropriate, in the absence of specific validation studies, some of the more speculative points about ISG function might be more appropriate in the Discussion section.

We agree with reviewer 1 that the results section contains some speculative points, which are normally discussed in the discussion section. However, we feel that we need to describe the data in the results section because the flow of the manuscript would otherwise not be logical for the reader. Coming back to the described results in the discussion section would similarly cause confusion. We do understand that this is not the classic way of writing a manuscript but is often used in screening manuscripts to explain the impact and relevance of the results shown within the section. We would appreciate if it was possible to keep this structure, alternatively, we suggest to add a “supplementary discussion” within the results section.

-Throughout the manuscript, certain terminology regarding what is measured/quantified in the microscopy data should be modified. For example, the authors refer to “viral spread” but unless

frequencies of infected cells were quantified by segmentation analysis or similar, this may not appropriately describe the data available.

We thank the reviewer for highlighting this important point regarding the terminology used in the manuscript. We carefully reviewed all instances of terms such as “quantified” and “measured” and revised them where appropriate. Specifically, to better reflect the microscopy analysis performed, we replaced the term “viral spread” with “virus propagation” throughout the manuscript, as it more accurately describes the methodology and data obtained.

-Relatedly, terminology such as “antiviral activities” implies direct effector functions. At Lines 127-129, counting STAT1 and IRF9 among ISGs with broad “antiviral activities” is therefore somewhat inexact; these are transcription factors that activate expression of all other ISGs (with antiviral activities). The same likely applies for DDX58 (Line 132), which likely acts as a viral sensor rather than effector.

We agree and carefully reviewed terminology throughout the manuscript. While it is clear that transcription factors have indirect roles, it is not clear to what extent other ISGs have direct or indirect activities. We explain the importance of pattern recognition receptors as regulators of the innate immune response in the introduction: “A notable subset of ISGs regulates the IFN system, like the pattern recognition receptors (PRRs) DDX58 (RIG-I), MDA5, and cGAS or the IFN signaling cascade, such as IRF1, STAT1/2, and IRF9, which are essential to establish an effective immune response” and now additionally added a paragraph describing in more detail the of PRRs and signaling molecules in the discussion section, which helps the reader to interpret the results.

-Line 51: “curtail” may not be the correct word here

We have replaced “curtail” with “established” to better align with the intended meaning and improve clarity.

-Line 71: use of “necessary” instead of “crucial” may align better with common terminology

We have replaced “crucial” with “necessary”.

Reviewer #2 (CRISPR LoF screen/systems)(Remarks to the Author):

The interferon (IFN) response is a highly potent anti-viral defense mechanism. Upon exposure of cells to IFN, hundreds of IFN stimulated genes (ISGs) are regulated, many of which known anti-viral factors. However, the virus species-specific impact of individual ISGs and their contribution to the overall anti-viral effect of IFN is still somewhat enigmatic. In this manuscript Krey and colleagues present a comprehensive overview of the contribution of ~300 ISGs to the anti-viral impact against 8 different viruses. They identify common anti/pro viral as well as virus-species specific ISGs and cluster the response patterns using machine learning. Finally, they functionally characterize two selected hits, RTCB and BORCS8.

Overall, this is a very impressive, highly interesting and relevant collection of data, which will certainly also inspire future compelling research. Generally speaking the novel conclusions of the manuscript are fully supported by the presented data. Especially ISG vs virus map and

corresponding analyses are very well planned and carried out, and provide significant insight in pan pro/anti-viral and specific pro/anti-viral impact of ISGs. However, the mechanistic analyses fall a bit short of explaining why the chosen candidate ISGs promote/restrict viruses and thus should be strengthened.

Major:

One important question is always, which ISGs are relevant for the IFN response against a specific virus i.e. what is a 'minimum' set of ISGs. Starting from the overview in Fig1 and Fig2 as well as the co-depletion analyses, can the authors define (I) which ISGs are likely to e.g. contribute 95% of the anti-viral effect of IFN α . (II) Calculate the contribution of synergistic active ISGs to the whole IFN-mediated restriction of SARS-CoV-2. Furthermore, could all machine learning neurons be clustered to a overarching principle or gene ontology term, and in the results section described which complexes are known, which are not (extending the current description)?

Reviewer 2 raises very interesting points. Unfortunately, they are very difficult to answer without overstatements for the following reasons:

(I) Defining a "minimum" set of ISGs contributing to 95% of the antiviral effect is challenging for various reasons: (a) Viruses are very diverse in their sensitivity towards ISGs. Some viruses may require cooperative activity of multiple ISGs to achieve 95% reduction, while other viruses may require fewer ISGs for the same inhibition. The limited dynamic range of our assay may compress individual effects and complicates direct comparisons to the overall antiviral response. (b) Potential differences in knockout efficacy could further influence the "strength" of individual ISGs, making absolute estimates difficult. (c) we show in Figure 4c that co-depletions of ISGs can have "neutral" effects. Individual depletion of STAT1 and IRF9 strongly increases growth of most viruses. However, double depletion does not show additive effects, as expected. Similar relationships could happen for any other combination of ISGs. For that reason, it is very difficult to define an absolute number/identity of top-ISGs required to confer protection.

As a solution, we provide Supplementary Table 2, where the activity of individual ISGs can be ranked by antiviral strength. This enables readers to select the top-active ISGs, which would be most useful information. However, we would like to refrain from giving exact "inhibition scores", if possible.

(II) Calculating the exact contribution of synergistically active ISGs to the IFN-mediated restriction of SARS-CoV-2 is similarly constrained. While the co-depletion studies offered valuable insights into ISG interactions, the assay's dynamic range limits precise quantification. In some double KO experiments, the system became saturated, with infection progressing too rapidly to capture the full kinetics, further complicating conclusive overall synergy analyses. As for point (I) we provide a list of ISGs that showed highest antiviral activity (Supplementary Table 2). We propose that the cooperativity of those ISGs confers the antiviral activity observed after interferon treatment. We also want to raise that the interferon response is cell type specific and the absolute abundance of ISGs may contribute differentially to the overall antiviral response.

(III) The machine-learning-based categorization of ISG activities is highly useful to identify proteins with similar functionality, but for reasons listed below, it is not possible to perform the requested enrichment analysis. First, for correct enrichment analysis, the number of overall tested proteins has to be considered, which is in our hands 285. This is a relative low number that does not reflect

a representative cross-section of possible biological functions. Also, all ISGs are functionally related (belonging to “antiviral” proteins), yet their individual function is often not clarified. This is particularly evident for relatively uncharacterized proteins (e.g. FAM-proteins). Another complication is the low number of proteins detected in some neurons, which prevents robust statistical enrichment analysis and therefore limits the strength of any conclusions.

While the functional analyses of RTCB and BORCS8 provide a lot of detail about their interaction partners, the molecular mechanism how they impact the respective viruses is still unclear. I suggest that the authors add experimental evidence of the impact of one of these factors on viral replication to strengthen the manuscript. For example, does depletion of BORCS8 increase fusion/entry of incoming virions (as the pseudotype assays would hint at)? Does ORF3a expression alone alter BORCS8 localisation patterns, hinting at a hijacking of BORCS8 by the viral accessory protein?

To address the potential role of BORCS8 in viral entry, we performed additional experiments that further support the hypothesis of enhanced endosomal entry upon BORCS8 depletion. Specifically, we tested the effect of Camostat, a TMPRSS2 inhibitor, in BORCS8 KO cells expressing TMPRSS2. While Camostat effectively reverted the proviral effect of TMPRSS2 expression alone, it had no impact on viral replication in BORCS8 KO cells. This indicates that in the absence of BORCS8, SARS-CoV-2 preferentially utilizes the endosomal entry route, which is independent of TMPRSS2-mediated membrane fusion. Together with the VSV-Spike pseudotype assay, which also showed increased viral entry in BORCS8 KO cells, these results provide strong evidence that BORCS8 depletion enhances endosomal viral entry.

ORF3a has been reported to alter localization of lysosomal proteins – such as LAMP1 – through hijacking lysosomal exocytosis (Chen et al. (10.1016/j.devcel.2021.10.006)). In line with the overall effect on lysosomal exocytosis we hypothesize that ORF3a likely influences the broader BORC complex rather than BORCS8 individually.

Minor:

Any virus-assay based solely on cell line data usually faces the criticism, whether the results would be the same in a more relevant setting i.e. primary cells. While I do understand that the main focus of this manuscript is to provide an ‘atlas’, it would inspire more confidence in the screen if a few selected (extreme) hit from the groups of pan pro/anti-viral and specific pro/anti-viral were confirmed in primary human lung cells.

We appreciate the reviewer’s suggestion and fully agree that validation in primary human cells strengthen confidence in the findings. We validated the activity of a subset of ISGs on primary Human Foreskin fibroblasts (HFF) and the neuroblastoma cell line SKBSH for three different viruses. We got identical KO phenotypes for infection with HSV-1 and for the other viruses, the majority of KO cells tested also showed results that were comparable to A549s. However, we also found that depletion of some proteins showed phenotypes that were deviating from the original observation in A549 cells. This is to some extent expected, since the expression values of ISGs and virus replication efficacy in different cell types can vary. The knockdown in different cell types is now added as additional results in Supplementary Figure 2b and further discussed in the discussion section in the manuscript.

One bottleneck of CRISPR screening approaches is always the efficiency of the gRNA. Can the authors briefly comment (in the results section) on how the (pools of) gRNAs were selected and whether/how the KO efficiency in A549 was quantified to allow the conclusion that each gRNA targets each ISGs efficient enough.

We now added additional descriptions of the gRNA selection into the results section: “gRNAs were selected based on high on-target scores, low off-target potential (avoiding predicted off-targets with ≤ 2 mismatches), and low self-complementarity to minimize undesired interactions. Pools targeting different regions of each ISG were used to enhance knockout efficiency. To assess knockout efficiency, we performed RT-qPCR on a subset of ISGs, showing substantial transcript reductions (Supplementary Figure 1d). Proteomics data (Supplementary Figure 3c) further confirmed decreased protein levels validating effective gene disruption. We have added these details to the Results and Discussion sections, acknowledging potential heterogeneity in knockout efficiency and its impact on phenotypic comparisons.

The authors may want to revisit their statements about TRIM25 and ZC3HAV1, also known as ZAP. TRIM25 is a rather well-known co-factor of ZAP, thus the results of the authors do not uncover a novel connection but rather confirm a well-known one within their data set. Compare: Li et al, Plos Path, 2017; Zheng et al, JVI, 2017; Nchioua et al, mBio, 2020.

We thank the reviewer for pointing out this oversight. We revised the manuscript accordingly.

line 400: ORF3a ?

The typo has been corrected.

Reviewer #3 (Innate immunity)(Remarks to the Author):

In this study, the authors systematically investigated 285 ISGs for their virus-modulating activity against eight viruses by utilizing a time resolved, arrayed loss-of-function screen. This comprehensive analysis identified several ISGs with yet unreported virus specificity and ISGs with pan-antiviral activity, such as RNA 2',3'-cyclic phosphate, 5'-OH ligase (RTCB). Furthermore, the authors identified BORCS8 which had a particularly prominent role in modulating SARS-CoV-2 infection by mediating the acidification of early endosomes during the viral entry, a process known to facilitate virus particle degradation. Overall, this study is potentially interesting. However, the novelty and depth of research need to be improved. Most of the findings about antiviral activity of various ISGs have been previously reported. BORCS8 as a regulator of SARS-COV-2 infection is only supported by in vitro studies. Other concerns are listed below.

To our knowledge, this is the first KO screen that tests the activity of this number of ISGs for multiple viruses in parallel. While the antiviral activity for a subset of ISGs had been reported before, and is recapitulated in this manuscript, a systematic comparison between different ISGs and different viruses had not been done. Moreover, for the vast majority of ISGs no loss-of-function studies have been reported to date. Since there was no data available, we used a systematic, time-resolved loss-of-function screening approach that consolidates known findings and uncovered many yet unreported virus-specific and pan-viral activities of ISGs. There are numerous surprising

functions, including many anti- and proviral activities. Most prominent examples are the pan-proviral role of RTCB and unreported specificities across multiple ISGs that are described in the results section. We thus believe that many novel activities of ISGs are presented within this manuscript and that this resource will serve as a foundation for further mechanistic studies and that many scientists would be interested in this manuscript.

1. The authors used only one cell line (A549) to evaluate the antiviral activity of the tested ISGs on various viruses, including Herpes simplex virus 1 (HSV-1) strain 17+, Measles virus (MeV) strain EdTag, Rift Valley Fever Virus (RVFV) ZH548 strain, Severe acute respiratory syndrome coronavirus 2 (SARS-CoV-2) MUC-IMB-1 strain, Semliki Forest virus (SFV) SFV6 strain, Vaccinia virus (VACV) WR strain (V300), Vesicular stomatitis virus (VSV) Indiana strain, and Yellow Fever virus (YFV) 17D strain. Different types of cells should be tested to confirm their findings.

A549 cells are widely utilized in antiviral research due to their well-characterized responsiveness to Type I interferons and their suitability for CRISPR-Cas9 KO studies. Their use in our study ensures robust reproducibility and comparability of results.

We agree with reviewer 3 that the use of additional cell types is interesting. Therefore, we further assessed the activity of a subset of selected ISGs in primary Human Foreskin Fibroblasts (HFFs) and in the neuroblastoma cell line SK-N-SH across three different viruses (HSV-1, VACV and RVFV). For HSV-1, the KO phenotypes were consistent with those observed in A549 cells, and for the other viruses most tested KOs showed comparable effects. Nevertheless, for some of the tested ISGs, depletion resulted in phenotypes that differed from our initial findings in A549s. This is not unexpected, as ISG expression levels and viral growth rates differ across cell types. This additional KO data using alternative cell types is now included in Supplementary Figure 2b and results are further discussed in the manuscript's Discussion section.

2. Most of the findings in this study is data-driving without experimental examinations. For example, in Figure 4, the synergistic effects of several ISGs, such as STAT2, LY6E, BORCS8 et al., should be confirmed by experiments in double- or triple-knockout cells.

We respectfully disagree with the reviewer's claim that most of the findings in our study are data-driven without experimental validation. The synergistic effects of ISGs were experimentally determined by generating more than 400 double-KO cells. The interaction scores were derived from experimentally measured infection kinetics, comparing observed double KO effects with the expected multiplicative outcomes of individual KOs. These results are based entirely on empirical data, as detailed in the Methods section. We regret any misunderstanding and hope this clarification addresses the reviewer's concerns.

3. The authors could introduce other databases such as Biogrid and String to analysis intercoms of candidate ISGs, which might provide additional insights for understanding how these ISGs function individually or synergistically.

We appreciate the reviewer's suggestion and mapped ISGs to STRING, as suggested. However, by doing so, we realized that there were several issues with this analysis, since many of the functionally relevant ISGs did not have annotated binding partners. Moreover, some ISGs change their interaction partners in interferon stimulated conditions. For instance, the interactome of IFIT

proteins is fundamentally different in conditions of interferon stimulation vs non-stimulated conditions. Both points are problematic since they would directly influence the interpretation of the resulting network. To overcome these limitations, we performed proteomics and interactome analysis of the ISGs showing the strongest anti-anti-SARS-CoV-2 activity. These experiments were done in absence and presence of interferon and virus infection and presented as Supplementary Figure 3. We are currently in the process to generate a systematic ISG-interactome with the functionally relevant proteins. However, we ask for the understanding of reviewer 3 that this work will be published as a separate manuscript.

4. In Figure 5, the authors found that BORCS8 mediated the acidification of early endosomes to regulate SARS-COV2 infection, which should be confirmed by more experimental evidences. Furthermore, the molecular mechanisms need to be explored.

Our study demonstrates that BORCS8 depletion enhances SARS-CoV-2 replication, likely by altering endosomal acidification and trafficking. We demonstrated that BORCS8 interacts with components of the BORC and BLOC complexes, which are crucial for lysosomal positioning and function, and that its depletion results in increased lysotracker and pHrodo signals, indicating enhanced endosomal acidification and cargo retention. Functionally, BORCS8 KO cells exhibited improved infection with SARS-CoV-2 and spike-pseudotyped VSV, supporting a role for BORCS8 in regulating viral entry. To further validate this, we performed rescue experiments by overexpressing a codon-optimized BORCS8 construct in BORCS8 KO cells, which reversed the enhanced viral replication phenotype. Most importantly, we expressed TMPRSS2 in A549 cells, which enables fusion of SARS-CoV-2 at the plasma membrane. Notably, under these conditions, depletion of BORCS8 did not influence SARS-CoV-2 growth. Inhibition of TMPRSS2 by Camostat, however, rescued the BORCS8 phenotype in BORCS8 KO cells. These results reinforce the conclusion that BORCS8 regulates SARS-CoV-2 infection by modulating endosomal acidification and trafficking. The new results are shown now in Figure 5e and discussed accordingly in the manuscript.

Reviewer #1 (Remarks to the Author):

This revised manuscript, “The ISG Atlas: A Loss-Of-Function Analysis Characterizes Antiviral Properties of Interferon Stimulated Genes,” from Krey et al represents a significant improvement from the initial submission. In general, the authors were responsive to reviewer comments, particularly with regard to mechanistic and follow up experiments involving BORCS8. Overall, as noted in the initial review and now further improved upon in the present submission, this manuscript remains highly significant and an impressive body of work. It is likely to be of considerable interest to the virology and innate immunity fields. As previously acknowledged, the strengths of the study remain the innovative experimental design (testing ISG loss of function in the context of an IFN response), the breadth of different viruses and ISGs evaluated, and the longitudinal timepoints to assess viral restriction. However, with regards to the ISG screen, the central focus of the manuscript, concerns remain about data interpretation and presentation, particularly with regard some of the quantitative comparisons made throughout the manuscript.

To state once again, the ISG/virus screen is very impressive and represents an important advance. However, it is clear both from the updated text and the responses to initial review that the screen (necessarily) includes many sources of heterogeneity (detailed below) across ISG and virus conditions. This raises significant concerns about how the data are interpreted and described throughout the manuscript; the authors frequently refer to certain ISGs as “more antiviral” than others, or ascribe “strong” pro/antiviral activities to certain ISGs, which may not be appropriate given the (understandable) technical limitations of the approach.

We acknowledge the concern of Reviewer 1. Heterogeneity is an inherent issue of a dataset of this scale, and we did our best to reduce heterogeneity wherever possible. We also added additional points to the discussion whereby making the readers aware of potential sources of heterogeneity.

There exist several sources of technical heterogeneity in the experimental design of the ISG screen:

-In a welcome response to the initial review, the authors provided qPCR data for CRISPR knockout efficiency for a subset of target ISGs. These data, presented in Supplemental Figure 1, demonstrate CRISPR-mediated knockout for all ISGs tested. However, the results also further emphasize that the knockout efficiency is considerably variable across ISGs.

Knockout efficiency measured by qPCR are by nature variable and not comparable between different ISGs. We provide „fold-change“ differences between NTC controls and ISG KO cells. This „fold-change“ difference is directly influenced by the expression value of the ISG (a highly expressed ISG will likely result in a higher „fold change“ in the knockdown). Therefore, we would expect differences in ISG knockout efficacy when comparing different ISGs.

-In an additional response to initial review, the authors helpfully clarified the normalization approach used in image analysis of the screen; specifically, “The normalization in our analysis was performed per image by calculating the ratio of viral fluorescence to host cell fluorescence, which we found to be most robust and led to the most reproducible results.” This seems an appropriate strategy, particularly for an arrayed screen of this size (the authors note that

thousands of images were analyzed). However, it introduces an additional source of variance that is likely to vary ISG-to-ISG. Whole-image fluorescence intensity does not necessarily represent “viral propagation.” One can imagine a single infected cell that increases fluorescence intensity over time. This might provide a similar signal to one cell maintaining fluorescence intensity but spreading infection to many adjacent cells. While not necessarily critical in designating an ISG as “anti” or “pro” viral, these differences could have implications for the mode of viral inhibition. Responsive to the initial review, the authors appropriately removed the descriptor “spread” from the manuscript (which cannot be evaluated without measure infection at single cell resolution). However, without a measure of infectious viral particles, “propagation” is similarly inappropriate (e.g. a non-propagating infected cell could increase fluorescence intensity over time).

We started to set up image analyses that enable us to compare the spread and intensity of the viral signal in the individual images using graph-based approaches. However, this is quite complex, and the size of the dataset (more than 700.000 images) further slowed down the process. However, we agree with reviewer 1 that such additional analysis could shed further light on the anti- and proviral activity of ISGs. For instance, one could imagine that the infection pattern within an ISG KO population would be indicative of a certain function of an ISG. We are considering this analysis for the future development of this project. As requested by the reviewer we largely removed „propagation“ and replaced it with other phrases indicating virus growth.

- Infections were performed at low and high MOIs, depending on virus. This is an appropriate design choice, given differences in virus replication rate, cytopathogenicity, and fluorescence intensities. However, it again makes it challenging to compare ISGs quantitatively, further compounded by the fluorescence normalization strategy highlighted above.

Our screen was set up to obtain the maximum signal-to-noise ratio for every individual virus tested. In this way we expected to identify the ISGs with the strongest activities. This enables us to perform quantitative analysis of different ISGs that are active against an individual virus. It also allows qualitative comparisons of viruses. However, the nature of the pathogens themselves makes quantitative comparisons between them difficult. Again, this is inherent to any screen of this kind.

In summary, the above sources of variation make quantitative comparisons of ISGs particularly challenging. To be abundantly clear, these technical limitations do not invalidate a very impressive and important collection of screens; as noted previously this is a very impressive and informative body of work and should be recognized as such! The ISG effects, particularly the antiviral effects, detected appear robust and indicative of important viral restriction functions. However, the authors are encouraged to be extremely cautious with their interpretations and related language regarding quantitative comparisons of ISG activities. It is appreciated that, appropriately, many of the ISGs are described as “broadly” acting or similar, which is appropriate and non-quantitative. However, this is not universal throughout the manuscript. And while it is further appreciated that the authors added some language describing potential variation of knockout efficiency to the Discussion section (lines 497-502), there remains insufficient discussion of how this and the above-described issues should affect interpretation of the data.

We added additional points in the discussion section of the manuscript discussing the points raised by reviewer 1.

In addition, related to the above: if K and tau values can be impacted by the highlighted sources of variation (particularly CRISPR knockout efficiency), how might using these values as input to the SOM impact results and interpretation?

The SOM integrates all data obtained in the dataset and ISGs with known similarities (e.g. ISGs important for IFN signaling) were classified in similar neurons. Given this benchmarking, we are positive that the SOM interpreted viral growth behavior correctly.

An additional consideration in the screen design is the potential impact of cell growth rates. The authors are quite rigorous with their assessment of impacts of ISG knockout on cell growth rates. However, it seems that the effects of knockout on cell growth were tested in the absence of IFN. This could be problematic because 1) Experiments were conducted in the presence of IFN. Experiments were conducted across 48-72 hours, which is more than enough time for >1-2 replicative cycles of A549 cells 2) The knocked out genes are ISGs, so they may not be expressed in the absence of IFN / potential effects on proliferation not apparent in absence of IFN, 3) IFN is known to affect cell proliferation (as acknowledged in the manuscript text). The authors should address this point and how it was managed in screen analyses.

It is common practice in such a screen to test cell growth in a given condition. It is difficult to mimic the variety of conditions that can occur and where the tested ISG may play a role. The antiviral and inflammatory response caused infections could affect growth of ISG KO cells. It may be very interesting to study the effects of IFNs, Interleukins, Chemokines on ISG growth – but this would be a manuscript by itself since it would likely result in a similar large dataset. From the data presented in our manuscript, we cannot make firm statements on the involvement of ISGs in cell growth under IFN/antiviral conditions. During infections we cannot discriminate between a potential negative effect of a gene depletion (+ induced inflammation/antiviral response) from cytopathic effects caused by viruses. We added this to the discussion.

Additional minor points:

Lines 48-49: “To maximize the efficacy of the cellular defense system, the host has expanded its repertoire of ISGs,” This sentence is unclear. Do the authors mean evolutionarily expanded? If so, clarifications (and citations would be appropriate).

We changed the sentence into: To maximize the efficacy of the cellular defense system, the host has evolved a large repertoire of ISGs with diverse functions, resulting in a broad array of tools to effectively control a wide diversity of viruses.

In the Supplementary Table 2 “categorized data” tab, category labels are rather confusing. With STAT1 and YFV as a “simple” example, the K_category is “2_major_proviral” (K = 8.593) which suggests that KNOCKOUT of STAT1 PROMOTES YFV (“proviral”) but tau_category is “2_major_slower” (tau = 6.227) which suggests that KNOCKOUT of STAT1 SLOWS YFV, but this is not correct...knockout of YFV promotes the speed of YFV as indicated on the Fig 1 heatmap. As I interpret the category labels, the “directions” for K and tau categories are “inverted” to each

other.

Tau is a mathematical value that describes the time it takes to reach half-maximum virus growth. It is not related to the absolute increase of virus growth. For instance, depletion of a very potent ISG, such as LY6E in case of SARS-CoV-2 would enable increase of virus growth for longer, leading to higher K-values (maximum signal) but the time to reach K/2 would be considerably longer than for NTC. Tau values are particularly important for the characterization of ISG functions and the SOM approach.

BORCS8 expression was induced with Doxycycline. Were +/- doxycycline controls included in these experiments (as doxycycline can have effects on viral replication and/or cell proliferation)?

We have tested the effects of doxycycline on virus growth and could not see a negative impact at the concentrations used.

Figure 1 lists panels “a” and “c” but no panel “b”.

Apologies, corrected.

In the Results text, what was considered when designating “pro/anti” viral? K or tau? It is not entirely clear as presently written.

We considered K values, we improved clarity.

Reviewer #2 (Remarks to the Author):

The authors have addressed all my concerns. Thanks a lot and great work!

as the mediator for reviewer #3

Reviewer 3 had four major issues:

1. Only one cell line was used by the authors to characterise the impact of KO/viral infection - The authors reasoned why the cell line in question was used, but now also additionally provided data in a second cell type (even primary cells). From my point of view that comment has been addressed.

2. Data-driving without experimental examinations - I agree, the foundation of (empiric) scientific progress is the combination of data collection and data interpretation. Thus, collecting data without interpretation is practically meaningless, and provides no conceptual advance. However, I do not agree that the presented manuscript is such an empty collection of data. The screens were verified, combinatorial effects looked at and mechanistic experiments performed. In addition, the data is interpreted, set in context and provides -according to my opinion- a significant advance.

We thank you for your evaluation. We provide substantial interpretation that is still within what a reader can digest when studying this manuscript. Additionally, the data in this manuscript is fully transparent and should inspire data-driven approaches to further delineate the functions of ISGs and the antiviral response.

3. The authors should introduce other database – This was addressed by the authors. I would agree that a full integration of IFN stimulated/non stimulated datasets is highly interesting but beyond the scope of the current study.

4. More mechanistic analyses are required for BORCS8 – The authors have provided more mechanistic data. This was in line with the answers given to the other reviewers.

According to my opinion, the authors have addressed all the issues raised by reviewer 3 sufficiently, and the manuscript has significantly improved.